# Identification of SH2 Domain-Containing Protein 3C as a Novel, Putative Interactor of Dipeptidyl Peptidase 3

**DOI:** 10.3390/ijms241814178

**Published:** 2023-09-16

**Authors:** Mihaela Matovina, Ana Tomašić Paić, Sanja Tomić, Hrvoje Brkić, Lucija Horvat, Lea Barbarić, Vedrana Filić, Marija Pinterić, Snježana Jurić, Akmaral Kussayeva

**Affiliations:** 1Division of Organic Chemistry and Biochemistry, Ruđer Bošković Institute, Bijenička Cesta 54, 10000 Zagreb, Croatia; atomasic@irb.hr (A.T.P.); tomic@irb.hr (S.T.); lea.barbaric@irb.hr (L.B.); snjezana.juric1980@gmail.com (S.J.); kussayeva.akmaral@gmail.com (A.K.); 2Faculty of Medicine, Josip Juraj Strossmayer University of Osijek, 31000 Osijek, Croatia; hrvoje.brkic@mefos.hr; 3Division of Molecular Biology, Ruđer Bošković Institute, Bijenička Cesta 54, 10000 Zagreb, Croatia; lhorvat@irb.hr (L.H.); vfilic@irb.hr (V.F.); 4Division of Molecular Medicine, Ruđer Bošković Institute, Bijenička Cesta 54, 10000 Zagreb, Croatia; mpinter@irb.hr

**Keywords:** protein–protein interaction (PPI), DPP3, SH2D3C

## Abstract

Dipeptidyl peptidase 3 (DPP3) is a zinc-dependent exopeptidase with broad specificity for four to eight amino acid residue substrates. It has a role in the regulation of oxidative stress response NRF2–KEAP1 pathway through the interaction with KEAP1. We have conducted stable isotope labeling by amino acids in a cell culture coupled to mass spectrometry (SILAC-MS) interactome analysis of TRex HEK293T cells using DPP3 as bait and identified SH2 Domain-Containing Protein 3C (SH2D3C) as prey. SH2D3C is one of three members of a family of proteins that contain both the SH2 domain and a domain similar to guanine nucleotide exchange factor domains of Ras family GTPases (Ras GEF-like domain), named novel SH2-containing proteins (NSP). NSPs, including SH2D3C (NSP3), are adaptor proteins involved in the regulation of adhesion, migration, tissue organization, and immune response. We have shown that SH2D3C binds to DPP3 through its C-terminal Ras GEF-like domain, detected the colocalization of the proteins in living cells, and confirmed direct interaction in the cytosol and membrane ruffles. Computational analysis also confirmed the binding of the C-terminal domain of SH2D3C to DPP3, but the exact model could not be discerned. This is the first indication that DPP3 and SH2D3C are interacting partners, and further studies to elucidate the physiological significance of this interaction are on the way.

## 1. Introduction

Dipeptidyl-peptidase 3 (DPP3, Uniprot-Q9NY33) is a zinc metallopeptidase that sequentially cleaves off dipeptides from unsubstituted amino termini of 3 to 10 amino acid residues-long peptides in vitro, showing broad specificity for four to eight residue substrates. It is ubiquitously present in organisms from bacteria to humans and is found in almost all human tissues tested (https://www.proteinatlas.org/ENSG00000254986-DPP3/tissue, accessed on 11 November 2022) [1]. Based on its activity and ubiquitous presence, it is assumed that it plays a role in the final stages of protein turnover in the cell [2]. Based on its affinity and activity toward various bioactive peptides, it is assumed that it has a role in the regulation of blood pressure and pain, but its physiological role is still obscure. The proposed physiological roles of DPP3 were exclusively related to its peptidase activity until it was identified as one of the proteins that bind Kelch-like ECH-associated protein 1 (KEAP1, Uniprot-Q14145), the inhibitor of nuclear factor erythroid 2-related factor 2 (NFE2L2, NRF2, Uniprot-Q16236) transcription factor, one of the major regulators of oxidative stress response in cells. The binding of DPP3 to KEAP1 prevents ubiquitination and degradation of NRF2 and causes the translocation of newly synthesized NRF2 to the nucleus and activation of the transcription of numerous genes involved in the oxidative stress response controlled by NRF2 [3,4]. The proposed role of DPP3 in oxidative stress was confirmed by the study using DPP3 KO mice, in which the absence of DPP3 was associated with a decrease in NRF2 activity and increased oxidative stress, leading to bone loss due to increased activity of the osteoclasts [5]. In another study using KO mice, DPP KO mice were found to have increased levels of several angiotensin-related peptides, resulting in higher water uptake and elevated levels of reactive oxygen species in the kidney. Interestingly, the effects were stronger in male mice, indicating involvement of the endocrine system, which attenuates the effects of DPP3 depletion [6]. Apart from its role in the oxidative stress response, recent findings of increased levels of circulating DPP3 in the plasma of septic and cardiogenic shock patients with adverse outcomes have raised interest in the research of DPP3 physiology and pathophysiology [7,8]. Several high-throughput studies have identified additional proteins interacting with DPP3, and The Biological General Repository for Interaction Datasets (BioGRID, https://thebiogrid.org/115383/summary/homo-sapiens/dpp3.html, accessed on 5 June 2023) lists 34 interactors of human DPP3; however, the interaction with KEAP1 is the only one confirmed by several groups to date. In order to find novel interactors of DPP3 and gain more insight into its physiological role, the stable isotope labeling by amino acids in cell culture coupled to mass spectrometry (SILAC-MS) approach was used to search for novel interactors of DPP3 in TRex HEK293T cells stably expressing HA-DPP3 as bait, and SH2 domain-containing protein 3C (SH2D3C) was identified as prey in one of four replicates of SILAC-MS interactome analysis (unpublished results).

SH2D3C (UniProtKB-Q8N5H7) is one of three members of the family of proteins that contain both an SH2 domain and a domain similar to guanine nucleotide exchange factor domains of the Ras family GTPases (Ras GEF-like domain). Mouse SH2D3C binds phosphorylated EphB2; hence, it was named SHEP1 (SH2 domain-containing Eph receptor-binding protein 1). It also binds small Ras family GTPases R-Ras and Rap1A but shows no guanine nucleotide exchange activity [9]. The 78 kDa isoform of the same mouse protein was identified as the protein that interacts with breast cancer antiestrogen resistance protein 1 (BCAR1, Cas, and p130cas) and HEF1 proteins through the C-terminal region, and was named Chat (Cas/HEF1-associated signal transducer). The protein was phosphorylated by mitogen-activated protein kinase (MAPK) upon epidermal growth factor (EGF) stimuli, and EGF stimuli also caused the translocation of both Chat and p130Cas from the cytoplasm to the ruffling membrane, and its overexpression activated c-Jun N-terminal kinase (JNK) [10]. SH2D3C facilitates cell migration toward EGF and induces membrane ruffling, most likely through physical interaction with the p130Cas protein, which promotes the binding of the Crk protein to p130Cas [11]. SH2D3C (Chat-H) is an indispensable regulator of integrin-mediated adhesion and chemokine-induced tissue-specific migration of primary T-lymphocytes, acting upstream of the small GTPase Rap1. The plasma membrane association of Chat-H, most likely mediated through its unique N-terminal domain, was required for T-lymphocyte migration, as well as the interaction with Cas-L and Chat-H-mediated phosphorylation of Cas-L. However, the exact mechanisms of Chat-H-mediated Rap1 activation are unknown [12]. Human homologs of the family were found by searching the expression-tagged library (EST) for SH2-containing proteins. Three newly found proteins were named NSP1 (SH2D3CA), NSP2 (BCAR3), and NSP3 (SH2D3C). NSP2 and NSP3 were widely expressed, while NSP1 had expression restricted to several tissues. SH2D3C and the other protein from the family, BCAR3, have multiple splice variants that differ in the 5′ untranslated region and amino-terminal sequences. The expression of variants is most likely controlled by different promoters, and they might be differentially expressed in cells [13]. The NSP-family proteins bind to Cas-family proteins through their Ras GEF-like domains, creating modular signaling complexes. The crystal structures of the BCAR3 and SH2D3C–p130Cas complex show that the Ras GEF-like domains of BCAR3 and SH2D3C adopt an unusual, closed structure which renders them catalytically inactive, but also allows them to bind tightly to p130Cas. Residues important for the NSP3–BCAR3-Cas interaction are conserved in both families, which allows various combinations of high-affinity interactions between different members of the NSP and Cas families [14]. In one study using SH2D3C KO mice, defects were found in olfactory bulb development [15], while, in another study using SH2D3C KO mice, defects were found in marginal zone B cell function, most likely caused by their impaired mobility, which could be the result of impaired integrin signaling [16]. Recently, SH2D3C was found to be upregulated in rat cortical neurons after treatment with amyloid-ẞ-oligomers, and higher protein levels of SH2D3C were also detected in mice with Alzheimer’s disease compared with wild type. Overexpression of SH2D3C also resulted in neuronal death [17]. Although there is much evidence for the involvement of SH2D3C in cell migration, adhesion, tissue organization, and regulation of immune response, the exact mechanisms of its involvement in these processes have not been fully elucidated.

We have confirmed the DPP3–SH2D3C interaction by several low-throughput experimental methods and a molecular docking investigation. This is the first indication that SH2D3C might be the interactor of DPP3, and their interaction might represent a link between the regulation of oxidative stress response via the KEAP1–NRF2 pathway and the processes in which SH2D3C is involved, including cell adhesion, migration, and growth.

## 2. Results

### 2.1. Overexpressed DPP3 Protein Interacts with Isoforms 2 and 3 of SH2D3C

FLAG-tagged isoforms 2 and 3 of SH2D3C protein were transiently overexpressed in TRex HEK293T cells stably overexpressing HA-DPP3. TRex HEK293T cells stably transfected with empty pcDNA.TO.HA plasmid were used as a negative control. Anti-HA coimmunoprecipitation was performed with agarose-conjugated mouse monoclonal anti-HA antibody, and binding was analyzed by Western blot with rabbit anti-FLAG antibody. Overexpressed FLAG–KEAP1 was used as a positive control (Figure 1a and Appendix A). Both isoforms of the SH2D3C protein bind DPP3 (Figure 1b and Appendix A).

### 2.2. Both Isoforms 2 and 3 of SH2D3C, as Well as the C-Terminal Ras GEF-like Domain Alone, Interact with WT DPP3 and the Catalytically Inactive Variant E451A

GST pulldown was performed to further corroborate DPP3–SH2D3C interaction and to investigate whether the C-terminal Ras GEF-like domain, which is present in all isoforms, is responsible for the interaction with DPP3. WT DPP3 and the catalytically inactive E451A variant of DPP3 were expressed in E. coli as fusions with the GST protein. GST–DPP3, GST–DPP3–E451A, and GST (negative control) were bound to glutathione agarose beads and mixed with the lysates from HEK293T cells overexpressing isoforms 2 and 3 of the SH2D3C protein and C-terminal domain, respectively. The binding of all three proteins to both WT and the catalytically inactive mutant of DPP3 was confirmed (Figure 2a–c, Appendix A), which confirms that the binding is mediated via the C-terminal domain and is not dependent on the DPP3 enzymatic activity.

### 2.3. EGFP-DPP3 and SH2D3C-mCherry Colocalize in Cytosol, Nucleus, and Membrane Ruffles

Live-cell imaging was performed to visualize the subcellular localization of EGFP–DPP3 and SH2D3C–mCherry in NIH3T3 cells transiently transfected with appropriate expression vectors. Colocalization of mCherry–Keap1 and EGFP–DPP3 was used as a positive control and was primarily detected in the cytosol (Figure 3, upper panel). EGFP–DPP3 and SH2D3C–mCherry also colocalized in the cytosol, with a weak signal in the nucleus. However, colocalization was also observed in the membrane ruffles (Figure 3, lower panel).

### 2.4. DPP3 and SH2D3C Interact Specifically

To investigate whether SH2D3C and DPP3 interact specifically in living cells, we employed the bimolecular fluorescence complementation (BiFC) assay. Venus fusion combinations were constructed for both proteins and transfected in the NIH 3T3 cells with complementary fusion pairs. The fluorescence resulting from the complementation of two Venus fragments was detected in live cells expressing DPP3-VenusfN in combination with SH2D3C-VenusfC and VenusfC-SH2D3C, respectively (Figure 4). We have observed Venus fluorescence in the same compartments in which DPP3 and SH2D3C were found to colocalize (Figure 3). Interestingly, the high-intensity signal was observed in the membrane ruffles at the cell’s edge; the intensity of the signal in the membrane ruffles was much stronger when VenusfC was positioned at the C-terminus of SH2D3C. Therefore, the BiFC assay demonstrated the direct interaction of DPP3 and SH2D3C in the cytosol and in the membrane ruffles of live cells, while the BiFC signal in the nucleus was very weak.

### 2.5. SH2D3C Protein Does Not Show GEF Activity towards Small GTPase RRAS

To test whether the SH2D3C protein has GEF activity toward the small GTPase RRAS, a luminescence-based GEF assay was performed with purified proteins, RRAS-amino acids 27-196 (RRAS_27-196), p120GAP, SH2D3C-isoform 2 and DPP3. It was shown that RRAS_27-196 has GTPase activity, which was enhanced in the presence of p120GAP. We have not detected GEF activity of SH2D3C, nor was it stimulated by the presence of DPP3 (Figure 5). We used a relatively low concentration (1 µM) of DPP3 that showed weak GTPase activity (Figure 5). We assume that DPP3, although being >95% pure (Appendix A), is contaminated with bacterial GTPases. The concentration of SH2D3C was adjusted accordingly.

### 2.6. Several Models of Binding of DPP3 and C-Terminal Domain of SH2D3C Are Revealed by Protein Docking Analysis

At least 10 conformations were generated with each of the servers mentioned in the ‘Material and Methods’ section, ClusPro, Haddock, and GRAMM-X. The best docking results were selected according to their docking score, and the results obtained by the different servers were compared and visualized. The overlap of the best-rated protein–protein docking results is shown in Appendix A. ClusPro mainly predicted the binding of SH2D3C to the lower domain of DPP3, sometimes also to the upper domain of DPP3 near the β-strands (yellow-colored regions in Figure 6). The Haddock and GRAMM-X results also predicted the binding of SH2D3C to the yellow-stained regions of DPP3, but also to the region near the ETGE loop of DPP3 (colored white in Figure 6). The complexes obtained by docking with different servers differed considerably, but their alignment according to DPP3 revealed that there are three preferred regions on the surface of DPP3 to which SH2D3C binds. The largest binding site, binding site 1, is located on the backside of DPP3, near the junction between the upper and lower domains. It includes regions in the upper domain, the loop between the two beta strands (amino acid residues 587–598), the α-helix 647–663 and the unstructured region 115–119, and the N-terminal region in the lower domain (Figure 6). The second binding site includes only regions in the lower domain and is defined by the α-helix 139–148, the most solvent-exposed segments of the downstream loop 149–155, and the C-terminal portion (714–723) of the protein. The third binding site is located near the ETGE loop (462–484) and two α-helices at the top of the upper domain (613–640). The amino acid numbers in DPP3 correspond to its canonical isoform 1 (Uniprot Q9NY33-1), while the amino acid numbers in SH2D3C correspond to isoform 2 (Uniprot Q8N5H7-2). This region was previously found to bind to the Kelch domain of the KEAP1 protein. It should be noted that the docked SH2D3C molecules bind to these regions in different orientations, even when they bind to the same region on the surface of DPP3. Depending on the orientation of SH2D3C, we selected the three most representative complexes in which binding site 1 of DPP3 is occupied (models 2–4), one representative complex in which SH2D3C is bound to binding site 2 (model 1), and one model (model 5) in which SH2D3C is bound to binding site 3.

When selecting the complexes for MD simulations, we also considered the mutual orientation of SH2D3C (NSP3) and p130cas in the crystallographically determined structure of their complex (PDB_ID: 3T6G) (Appendix A [14]). When we superimpose SH2D3C (NSP3) in this structure with SH2D3C in the DPP3–SH2D3C complexes obtained by docking, we can see that the helices of p130cas with which SH2D3C (NSP3) interacts overlap with several helices of the lower domain of DPP3 (Appendix A) in model 1, whereas in the other models, they overlap with some other DPP3 regions, either in the lower domain or in the region of the hinge.

Initially, all these models were simulated for 200 ns in water, and their MMGBSA energies were calculated throughout the MD simulations (Appendix A). For model 2, the MMGBSA energies are positive at all intervals, so it was not considered for further simulations. The four models with the lowest energies were simulated further, model 1 for an additional 200 ns and models 3, 4, and 5 for an additional 500 ns (Appendix A).

During the last 140 ns (from the 560th to the 700th ns) of the MD simulations, the orientation of SH2D3C changed significantly (Figure 7 and Appendix A) in model 4. That is, after 560 ns of the MD simulations, SH2D3C moved away from the lower domain of DPP3, and the strength of their interaction decreased, as can be seen from the values of the LIE energies. The binding free energy calculated by the MMGBSA method also became less favorable (Appendix A). Comparing the MM (ΔG_gas_) energy (change in the protein–protein interaction energy) and the GBSA energy (ΔG_sol_) (change of free energy of solvation), it can be seen that the former energy increased (from ca. −150 kcal/mol to values > 0) and the other decreased from values around 100 kcal/mol to values around −30 kcal/mol, indicating that after 560 ns of the MD simulations, the protein–protein interface decreased and they became more solvated. Since the change in interaction energy calculated for conformers generated after 560 ns is greater than the change in solvation energy, the free energy of binding for DPP3 has increased (Appendix A). The radius of gyration (Rgyr) of model 5 also increased slightly during this period. Interestingly, Rgyr is lowest in model 3, although the Rgyr of DPP3 is larger in this complex than in the other complexes (Appendix A). Namely, the DPP3 interdomain cleft is more open and parts of SH2D3C bind in it (Appendix A). The Rgyr of SH2D3C remained at about 19.4 Å in all models and increased to 19.7 Å only in model 4 during the last period of the MD simulations (Appendix A).

The binding free energies calculated using the LIE method indicated model 3 as the most stable (Appendix A), while the lowest MMGBSA energies were calculated for model 5 (Appendix A and Figure 8). The reason for the discrepancy in these energies is the solvation component of binding, i.e., the desolvation of proteins upon binding is much less favorable in the case of model 3 than in model 5 (Appendix A). Namely, the intermolecular interaction energies calculated by the MMGBSA approach (denoted as ‘free energy of binding in gas’) are lower for model 3 than for model 5, which is consistent with the LIE energies, but the electrostatic component of desolvation in the formation of the protein complex is much higher in the case of model 3 than in the case of model 5 (Appendix A). Hydrogen bonding analysis also shows that there are many more intermolecular interactions between DPP3 and SH2D3C in model 3 than in model 5 (Appendix A). The most populated polar intermolecular hydrogen bonds between DPP3 and SH2D3C in model 3 are R125-E562, R598-E613, R159-D684, and T112-E604, and in model 5, R624-E607, R620-E607, and Q484-Q474 (Appendix A and Figure 9). In addition to direct protein–protein interactions, there are also water-mediated intermolecular hydrogen bonds, particularly in model 3 (Figure 10). Since, in model 3, SH2D3C interacts with amino acid residues from both DPP3 domains, the upper and the lower, there is a lot of space in between that is filled by the water molecules that make polar contacts and hydrogen bonds with the amino acid residues of the interacting proteins; in some cases, the same water molecules interact with the amino acid residues of the two proteins and bridge their interaction (forming the bridge between them) (Appendix A).

## 3. Discussion

According to various estimates, the number of protein–protein interactions in the human proteome is about 650,000, most of which are still unknown [18]. There are many proteins for which no interactors are known and many proteins with less than five interactors. One of these proteins is DPP3, which has only one confirmed interactor, KEAP1 protein. DPP3 is a biochemically and structurally well-studied protein whose physiological role is still not fully illuminated. Apart from the roles based on its dipeptidyl cleaving activity, DPP3 is also involved in the regulation of the KEAP1–NRF2 oxidative response pathway through its interaction with KEAP1 [4,19]. Several high-throughput studies have identified additional proteins that interact with DPP3, and The Biological General Repository for Interaction Datasets (BioGRID, https://thebiogrid.org/115383/summary/homo-sapiens/dpp3.html, accessed on 7 June 2023) lists 34 interactors of human DPP3 [20]. Almost all of them were identified in high-throughput studies and none of the interactions (apart from the one with KEAP1) was confirmed. In another study, in which DPP3 served as bait, nine novel proteins were identified as prey, and none of them were analyzed further [19]. Recently, it was reported that DPP3 interacts with CDK1 protein in the colorectal cancer cell line and that DPP3 knockdown decreases proliferation, causes G2 arrest, promotes apoptosis, and inhibits cell migration. The results were confirmed in an animal model [21]. Our own SILAC-MS analysis of the DPP3 interactome in HEK293T cells yielded more than 200 hits in 4 replicates, with 31 proteins identified as prey in at least 2 replicates (unpublished results). We used co-IP to confirm the interaction of DPP3 with 10 selected proteins that had the highest SILAC-MS score, 3 of which were found in 3 replicates, but none of the interactions were confirmed. However, co-IP confirmed DPP3 interaction with the SH2D3C protein, identified as prey in only one replicate of SILAC-MS. It is worth mentioning that the KEAP1 protein was also identified in only one replicate of SILAC-MS (unpublished results). SH2D3C has a very low mRNA expression in HEK293 cells (https://www.proteinatlas.org/ENSG00000095370-SH2D3C/cell+line, accessed on 7 June 2023) [1]; we cannot detect the protein in HEK293T cells by Western blot, nor were we able to amplify the SH2D3C cDNA insertion for cloning cDNA transcribed from RNA isolated from HEK293T cells, implying that SH2D3C expression in the cells used to produce the DPP3 interactome is indeed very low. This was further evidence that SH2D3C may be a genuine interactor of DPP3 and the reason we decided to test this interaction using multiple methods. The interaction of overexpressed isoforms 2 and 3 of SH2D3C with DPP3 was confirmed by co-IP (Figure 1, Appendix A) and GST-pulldown indicated that binding was not dependent on the DPP3 enzymatic activity and that SH2D3C binds DPP3 through its C-terminal Ras GEF-like domain (Figure 2, Appendix A), which is the same domain that binds p130Cas protein [17]. Colocalization of DPP3 and isoform 1 of SH2D3C was detected predominantly in the cytosol with weak staining in the nucleus, but we also detected colocalization in membrane ruffles (Figure 3). These results were confirmed by a BiFC assay with DPP3 and isoform 1 of SH2D3C (Figure 4). The finding that DPP3 and SH2D3C interact in membrane ruffles was another indication that this interaction might be physiological, since human DPP3 is a predominantly cytosolic enzyme and, to our knowledge, has not previously been detected in membrane ruffles, although there are reports of its localization in the membrane in calf and rat brains and several other rat tissues [22]. Translocation of DPP3 to the nucleus in response to hyperoxia that was augmented by estrogen treatment was also reported [23], as well as its presence in extracellular fluids, despite lacking any exporting signals in the amino acid sequence. The presence and activity of DPP3 in plasma have recently been identified as a biomarker of adverse outcomes in patients suffering from cardiac shock, but it has also been detected in the plasma of healthy individuals [7]. On the other hand, several studies have reported localization of SH2D3C in the membrane ruffles. The mouse homolog of SH2D3C (which has more than 85% identity with the human homolog), also known as CHAT and SHEP1, translocates to the membrane ruffles where it colocalizes with p130Cas in the EGF-stimulated COS7 cells [24]. These results were corroborated by another study that also showed that the effect of SH2D3C on membrane ruffling is mediated through its interaction with the p130Cas protein and that both proteins are detected in the membrane ruffles [11]. Our BiFC results (Figure 4), showing that the interaction of DPP3 and SH2D3C is localized in the membrane ruffles, could be the indication of its involvement in the control of cell migration. We have also tested if the binding of DPP3 to SH2D3C could induce its GEF activity toward small GTPase RRAS since SH2D3C has Ras GEF-like domain, which is inactive because it adopts closed conformation compared to the active GEF domains [17]. SH2D3C did not show Ras GEF activity toward RRAS, neither alone nor in the presence of an equimolar concentration of DPP3; however, we have shown for the first time that p120GAP is GTPase activating protein (GAP) for RRAS (Figure 5). P120GAP was identified as a GAP for the Ras family of small GTPases [25]. However, based on the literature search, no one has previously shown that it acts on RRAS.

Confirmation that the C-terminal Ras GEF-like domain also binds DPP3 was important for our molecular docking studies because only the crystal structure of this domain is deposited in PDB (PDB_ID: 3T6G). Molecular docking of the C-terminal domain of SH2D3C and DPP3 revealed five potential models of DPP3–SH2D3C complexes with SH2D3C binding in three different locations on the DPP3 surface (Figure 6). The first putative binding site is located on the backside of DPP3, near the junction between the upper and lower domains (model 1). The second binding site involves only regions in the lower domain, and SH2D3C could bind to this site in three different orientations (models 2–4). The third binding site is located near the ETGE loop and is potentially the most interesting, as this is the region through which DPP3 interacts with KEAP1. Model 2 was excluded from further analysis, but four other models are plausible, with model 3 being the most stable and model 5 having the lowest MMGBSA energies (Figure 8). It is difficult to determine the influence of the particular mode of DPP3–SH2D3C interactions on their function because the full structure of SH2D3C is not available, only the structure of its Ras–GEF-like domain. It should be noted that Alphafold structure prediction indicates that SH2D3C has a relatively high content of intrinsically disordered regions (IDRs) (more than 40%), [26] and we have modeled only the binding of one structured domain to DPP3, which may be one of the reasons why we obtained several similarly plausible binding modes. We might assume that the interactions between DPP3 and SH2D3C observed in model 5 and the structural dynamics significantly affect the interaction of DPP3 with the KEAP1 protein because, in this model, DPP3 interacts with SH2D3C through the same region as with the Kelch domain of KEAP1. On the other hand, the region through which DPP3 interacts with SH2D3C in model 3 is far from the region through which it interacts with KEAP1, so DPP3 would be able to interact with both proteins simultaneously in this case. However, this binding mode could affect enzyme activity because it involves regions of both DPP3 domains and, in this way, rigidifies the long-range dynamics of the protein, which are described as movements between domains (closing and opening) and are important for substrate binding and product release.

Proteins with a high content of IDRs are more flexible than proteins with a defined tertiary structure and, due to their structural flexibility, are often involved in connecting different cell signaling pathways and act as signaling hubs [27]. Based on the current knowledge of DPP3 and SH2D3C and our own findings, we propose that their interaction represents a link between NRF2–KEAP1 mediated oxidative stress response and the regulation of cell migration through the NSP–Cas network. Currently, there is no evidence that these signaling pathways are related; however, there are reports that DPP3 depletion by shRNA reduces the cell migration of colon cancer cell lines HCT-116 and RKO [18] and esophageal cancer cell lines Eca-109 and TE-1 [28], although the exact mechanism of DPP3 involvement in cell migration was not elucidated. In addition, both DPP3 and p130Cas (interactor of SH2D3C) have been linked to bone development through the regulation of osteoclasts. DPP3 knockout increased bone resorption by osteoclasts through the inhibition of the NRF2–KEAP1 pathway and a consequential increase of oxidative stress [5], while p130Cas acts oppositely; it is essential for the activation of osteoclasts and acts downstream of c-Src by controlling the small GTPase Rac1 activity [29]. Our investigations of DPP3–SH2D3C interaction on endogenous proteins have been hampered by the lack or low expression of SH2D3C in commonly used cell lines and the lack of good antibodies for IP and immunofluorescence. However, we think that the results obtained by analyzing the interaction between overexpressed proteins and molecular docking studies provide strong evidence for the validity of the interaction.

## 4. Materials and Methods

### 4.1. Experimental Study

#### 4.1.1. Cell Culture and Transfection

Adherent cell lines (HEK293T, NIH3T3) were maintained in a complete high glucose (4.5 g/L) Dulbecco’s Modified Eagle’s medium (DMEM, Sigma-Aldrich, Saint Louis, MO, USA) supplemented with 10% fetal bovine serum (FBS), 1% nonessential amino acids, and 1% antibiotic/antimycotic solution (all chemicals were purchased by Capricorn Scientific GmbH, Ebsdorfergrund, Germany). The cells were kept at 37 °C in 5% CO_2_ in a HeraCell 150 humidified incubator (Heraeus, Hanau, Germany). Cells were counted on a LUNA-II Automated Cell Counter (Logos Biosystems, Dongan-gu Anyang-si, Gyeonggi-do, South Korea). TRex HEK293T cells stably overexpressing HA-DPP3 for coimmunoprecipitation (co-IP) and HEK293T cells for GST pulldown were seeded in 6-well plates at 1 × 105 cells per well, grown until they were 70–90% confluent, and then transfected with Lipofectamine 2000 (Thermo Fisher Scientific, Waltham, MA, USA) and expression plasmids in 1 µg DNA: 2.5 µL Lipofectamine 2000 or 2 µg DNA: 5 µL Lipofectamine 2000 ratio and grown for 24–48 h before harvesting and isolation of the proteins. TRex HEK293T cells for co-IP were also treated with 1.5 µg/mL doxycycline for 24 h before the harvest to induce the expression of HA-DPP3. Embryonic mouse fibroblast (NIH 3T3) was used for confocal microscopy experiments. Cells were seeded in 4-chamber 35 mm glass bottom dishes (IBL Baustoff+Labor GmbH, Gerasdorf bei Wien, Austria) at 8 × 10^3^ cells per well. The suitable cell confluency at the time of transfection was 35–50%. Cells were transiently transfected with pEGFP–DPP3 and SH2D3C–mCherry or mCherry–Keap1, respectively, for colocalization experiments or with Venus-based BiFC constructs (for BiFC) using Lipofectamine™ LTX Reagent with PLUS™ Reagent (ThermoFischer Scientific), also according to the LTX instruction manual. The ratio of DNA (µg) to Lipofectamine 2000 or LTX (µL) was 1:2.

#### 4.1.2. Cloning

The cDNA insert for the expression of human DPP3 was amplified by PCR from plasmid pET21b–hDPP3 and cloned into the XhoI site of pcDNA4.TO eukaryotic expression vector with N-terminal HA-tag (pcDNA4.TO.HA). Plasmids were used to prepare stably transfected, inducible T-Rex HEK293T cell lines that express wild-type (WT) DPP3.

The cDNA insert for cloning of SH2D3C isoform 1 was amplified from the plasmid retrieved from human ORFeome V5.1 cDNA collection (Plate 31039, Internal ID: 9614), while isoform 2 was amplified from a different clone of hORFeome V5.1 (Plate 11071, Internal ID: 8299). Plasmids from the hORFeome collection are a kind gift from Dr. Oliver Vugrek (Ruđer Bošković Institute, Zagreb, Croatia). The cDNA inserts for SH2D3C-isoform 3 and the C-terminal domain (amino acids G539-L860 of the longest isoform 1) were amplified from the vectors with cloned SH2D3C-isoform 1. DPP3 cDNA was cloned in the pGEX-6P1 vector for the expression of GST-DPP3. Catalytically inactive E451A mutant was created by a QuikChange II XL Site-Directed Mutagenesis Kit (Agilent). All cloning was done by the use of the In-Fusion HD Cloning Kit (Takara Bio).

The DPP3 cDNA insert was cloned into the pEGFP-C1 vector between the XhoI and PstI site and the KEAP1 cDNA insert was cloned in the pmCherry-C1 vector between the XhoI and PstI restriction sites by restriction cloning. SH2D3C-isoform 1 was cloned into the mCherry-C1 vector upstream from mCherry by recombination cloning using the In-Fusion HD Cloning Kit (Takara Bio).

pcDNA3.1-VenusfN and pcDNA3.1-VenusfC vectors were prepared by In-Fusion (Takara Bio) recombination cloning of N-terminal 158 amino acids-long Venus fragment (Venus N) and the C-terminal 82 amino acids-long Venus fragment (VenusC), respectively, amplified from the Gateway vectors kindly provided by Dr. Oliver Vugrek (Ruđer Bošković Institute, Croatia) [30], into pcDNA3.1. BiFC vectors were constructed to cover all four fusion topologies (pcDNA3.1–VenusNfN, pcDNA3.1–VenusNfC, pcDNA3.1–VenusCfN; and pcDNA3.1-VenusCfC). Genes of interest (DPP3 and SH2D3C isoform 1) were cloned in all 4 vectors with the In-Fusion HD Cloning Kit (Takara Bio).

The RRAS cDNA insert comprising amino acids 27-196 of RRAS protein (UniProtKB-P10301) was cloned in the pET15b vector with the In-Fusion HD Cloning Kit (Takara Bio).

Primers used for cloning are listed in Appendix A.

#### 4.1.3. CoImmunoprecipitation

PFLAG-CMV2 vectors expressing SH2D3C-isoforms 2 and 3, respectively, were transfected in TRex HEK293T cells stably expressing HA-DPP3. TRex HEK293T cells stably transfected with empty pcDNA4.TO.HA vector and transiently transfected with pFLAG-CMV2 with appropriate insert served as negative control. Cells were harvested 24–48 h post-transfection in lysis buffer (50 mM Tris, pH 7.5, 75 mM NaCl, 1 mM EDTA, 0.1% Triton X-100) with 1X HALT protease inhibitor cocktail (Thermo Scientific). To each reaction and negative control, 20 µL of mouse monoclonal anti-HA agarose beads (Sigma #A2095) was added and incubated at 4 °C for 2–3 h. After incubation and wash with lysis buffer, the proteins were eluted from the beads with the addition of 20 µL of 1X TB (62.5 mM Tris-HCl, pH 6.8, 2% SDS, 10% glycerol, 1% 2-Mercaptoethanol, and 0.05% Bromophenol blue) and heated to 70 °C for 15 min. Co-IP reactions were analyzed by Western blot with the rabbit anti-FLAG antibody (Sigma #F7425).

#### 4.1.4. GST-Pulldown

GST–DPP3, GST–DPP3–E451A (catalytically inactive mutant) and GST were expressed in BL21-CodonPlus(DE3)-RIL E. coli strain. Bacterial cells were lysed in PBS/1% Triton-X-100 and bound to glutathione–agarose beads (Thermo Scientific) at 4 °C overnight and then washed 3 times with PBS/1% Triton-X-100.

PFLAG-CMV2 vectors expressing SH2D3C isoforms 2, 3, and the C-terminal domain were transfected in HEK293T cells. Cells were harvested 24–48 h post-transfection in PBS/1% NP-40 with 1X HALT protease inhibitor cocktail (Thermo Scientific). Cell lysates were split into 3 parts and glutathione–agarose beads bound with GST-DPP3, GST-DPP3-E451A, and GST were added to the lysates, respectively. GST-pulldown reactions were incubated for 1 h at room temperature, washed with PBS/1% NP-40, and the proteins were eluted from the beads with the addition of 20 µL of 1X TB and heated to 98 °C for 5 min. GST-pulldown reactions were analyzed by Western blot with the rabbit anti-FLAG antibody.

#### 4.1.5. Confocal Microscopy

Fluorescence experiments in live-cell imaging were performed with a laser-scanning confocal microscope Leica TCS SP8X (Leica Microsystems, Wetzlar, Germany), using an HC PL APO CS2 63×/1.4 NA oil-immersion objective, a 405 nm diode laser, and a supercontinuum white-light laser. Images were processed in LAS X Leica Microsystems software packages using Photoshop CS5. The excitation wavelengths and detection ranges used for imaging were 514 nm and 520–560 nm for Venus, 488 nm and 500–540 nm for EGFP, and 587 nm and 600–640 nm for mCherry.

#### 4.1.6. Protein Purification

DPP3 protein with C-terminal HIS-tag was expressed and purified as previously described [31], with the additional purification step of size exclusion chromatography on HiLoad 16/60 Superdex 200 prep grade column performed on the Åkta FPLC system (GE Healthcare. Chicago, IL, USA). A shorter variant of RRAS (amino acids 27-196) was expressed in *E. coli* with N-terminal HIS-tag and affinity purified on ROTI^®^Garose-His/Co Beads (Carl Roth, Karlsruhe, Germany) columns. Both proteins were desalted into 25 mM Tris (pH = 7.5), 300 mM NaCl, 1 mM DTT, 1 mM EDTA, and 10% glycerol buffer using Amicon Ultra-15 10K centrifugal filters (Merck Millipore, Burlington, MA, USA). Protein concentrations were determined using BioDrop for measuring protein A280 adjusted by their mass-extinction coefficient. Isoform 2 of the SH2D3C protein was expressed in the Baculovirus expression system in insect cells and purified in the EMBL Protein Expression and Purification Core Facility, Heidelberg, Germany.

#### 4.1.7. Guanine Nucleotide Exchange (GEF) Activity Test

The putative GEF activity of SH2D3C with or without DPP3 was analyzed using the GTPase-Glo Assay kit (Promega, Madison, WI, USA) according to the manufacturer’s instructions. To test SH2D3C GEF activity toward RRAS, GTPase activity of 2 μM RRAS was determined in the absence and presence of purified 1 µM SH2D3C-isoform 2 and purified 1 µM DPP3 in the GEF buffer containing 5 μM GTP and 1 µM p120GAP (BPS biosciences, San Diego, CA, USA) in 384-well, flat-bottom white plates (Greiner Bio-one, Kremsmünster, Austria). Reactions were incubated for 2 h at 24 °C; then, the GTP that remained in the solution was converted to ATP by nucleoside-diphosphate kinase and generated ATP was used in a luciferase reaction to produce light. Luminescence was measured on a multimode plate reader Spark (Tecan, Männedorf, Switzerland). As GTP hydrolysis is inversely proportional to the luminescence signal, the luminescence signal of the buffer control was set to 100%, and the luminescence of the rest of the reactions, including 2 and 4 µM RRAS, 2 µM RRAS with 1 µM p120GAP, 2 µM RRAS, 1 µM p120GAP and 1 µM SH2D3C, and 2 µM RRAS, 1 µM p120GAP, 1 µM SH2D3C, and 1 µM DPP3 was calculated as a percentage of luminescence of buffer control. Two additional control reactions were performed with only 1 µM SH2D3C and 1 µM DPP3, respectively. Three replicates of the experiment were performed, and the data was analyzed in GraphPad Prism 10 software. An unpaired, two-tailed *t*-test was used for statistical analysis.

### 4.2. Computational Study

#### 4.2.1. Protein Docking

For docking, we used the experimentally determined structure of the C-terminal (Ras–GEF-like) domain of the SH2 domain-containing 3C protein (SH2D3C; alt. names NSP3, SHEP1, CHAT) in complex with p130Cas (PDB_ID: 3T6G) and two different structures of the human DPP3, one crystallographic (‘open’ PDB_ID: 3FVY) and one semiopen, for which previous molecular simulations showed that is the most abundant in solution [32]. Since the crystallographically determined structure of the SH2D3C protein lacks 14 amino acid residues (residues in a sequence in the range 598–613), the indicated structure had to be modeled by homology, which was done with the program Modeller [33].

The prepared structures of DPP3 and SH2D3C were uploaded to several molecular docking servers capable of providing reliable results for large macromolecular systems [34] (ClusPro [35,36,37], Haddock [38,39], and GRAMM-X [40,41]), developed in Vakser’s group, and free docking was performed. The resulting structures of DPP3 and SH2D3C complexes were clustered, carefully examined, and compared.

#### 4.2.2. System Preparations

For molecular modeling, the protonation of the charged residues and histidines was adjusted to a pH of about 7.5, as expected under physiological conditions. Thus, the arginine and lysine residues were positively charged in our models, whereas the glutamate and aspartate residues were negatively charged. The protonation states of the histidines were determined using the H++ server [42] and additionally adjusted according to the results of our previous QM/MM studies [43] and their ability to form either hydrogen bonds or polar contacts with the surrounding residues. With the exception of H568 in DPP3 and H251 in SH2D3C, where both Nδ and Nε were protonated, all other His residues were neutral. All systems were parameterized with the ff19SB force field [44]. The zinc ion in the active site of DPP3 was described using the hybrid bonded–nonbonded model for metallopeptidases [45]. The system was solvated using a truncated octahedron of OPC-water molecules [46] which is recommended for use with the ff19SB force field [39]. The distance of the molecular surface from the box was at least 11 Å. Na+ (and/or Cl^−^) ions designed for use with this water model [47] were added to achieve electroneutrality or the desired salt concentration. All MD simulations were performed using the AMBER20 suite of programs [48,49] (https://ambermd.org/, accessed on 6 September 2021).

#### 4.2.3. Classical MD Simulations

Prior to the productive MD simulations, the systems were optimized in three cycles with different restraints. In the first cycle (1500 minimization steps), aimed at relaxing the solvent molecules, the protein and the zinc ions were constrained using a harmonic potential with a force constant of 32 kcal mol^−1^ Å^−1^. In the second cycle (3500 minimization steps), only the protein backbone was constrained with a force constant of 12 kcal mol^−1^ Å^−1^, while the entire system was minimized in the third cycle (2500 minimization steps) without additional restraints. The systems were heated from 0 to 300 K during the 25 ps-long MD simulations followed by a 3 ns long equilibration at 300 K. A time step of 0.5 fs was used for the heating simulations and 1 fs for the equilibration simulations.

In the productive MD simulations, we used the algorithm SHAKE [50] and a time step of 2 fs. During heating, the NVT ensemble was used, while equilibration and production MDs were performed with the NPT ensemble, with a cutoff value of 11 Å. During the simulations, the temperature was controlled using the Langevin thermostat [51] with a time interval between temperature rescaling of 0.5 ps during heating and density equilibration and of 1 ps during productive MD simulations. Pressure was controlled using the Berendsen barostat [52] with a relaxation time of 1.0 ps. A total of 2 μs of productive classical MD simulations were performed for the predicted DPP3–SH2D3C complexes.

#### 4.2.4. Data Analysis

Calculations of geometry parameters (RMSD, Rgyration, and RMSF), hydrogen bonding analysis, and analysis of LIE were performed using the cpptraj module [53] of the AmberTools20 program package and the Hbonds plugin of the VMD program (http://www.ks.uiuc.edu/Research/vmd, accessed on 7 March 2023) [54], with angle and distance cutoffs of 45 ° and 3 Å, respectively. The hydrogen bond population is reported as the ratio between the frames containing the bond and the total number of frames sampled during the simulations. MMGBSA energies were calculated at 10 ns intervals throughout the trajectory using the MMPBSA.py script implemented in the AMBER20 program suite [55]. An internal dielectric constant of 2 was used, as this has given good results in our previous work [55,56,57,58], while the ionic strength was set to 0.100 mol dm^−3^. Figures were generated using PyMOL (PyMOL Molecular Graphics System, version 1.5.0.4, Schrödinger LLC, New York, NY, USA).

## 5. Conclusions

We have identified a novel, putative interactor of the DPP3, SH2D3C protein. This is the first indication that these two proteins are interacting partners. Interaction between overexpressed proteins was confirmed by several methods and the computational analysis also supports binding of the C-terminal domain of SH2D3C to DPP3. Although our data does not fully confirm the existence of the physiological interaction between DPP3 and SH2D3C, the results obtained provide strong evidence for the validity of this interaction. Further investigations of the interaction between DPP3 and SH2D3C and its potential physiological role are on the way.

## Figures and Tables

**Figure 1 ijms-24-14178-f001:**
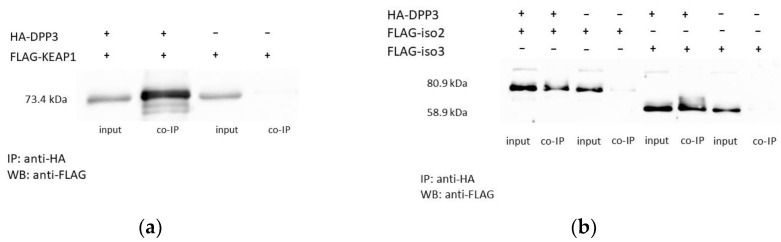
Western blot analysis of the co-immunoprecipitation (co-IP) experiments: (**a**) anti-HA antibody was used to immunoprecipitated HA-DPP3 from the lysate of HEK293T cell overexpressing HA-DPP3 and FLAG-KEAP1, and cells overexpressing only FLAG-KEAP1 (negative control); western blot analysis with anti-FLAG antibody was used to check if FLAG-KEAP1 is co-immunoprecipitated with HA-DPP3; (**b**) anti-HA antibody was used to immunoprecipitated HA-DPP3 from the lysate of HEK293T cell overexpressing HA-DPP3 and FLAG-SH2D3C-isofom 2 or 3, respectively, and cells overexpressing only FLAG- SH2D3C-isofom 2 or 3, respectively (negative controls); western blot analysis with anti-FLAG antibody was used to check if FLAG-SH2D3C-isofom 2 and 3, respectively, are co-immunoprecipitated with HA.

**Figure 2 ijms-24-14178-f002:**
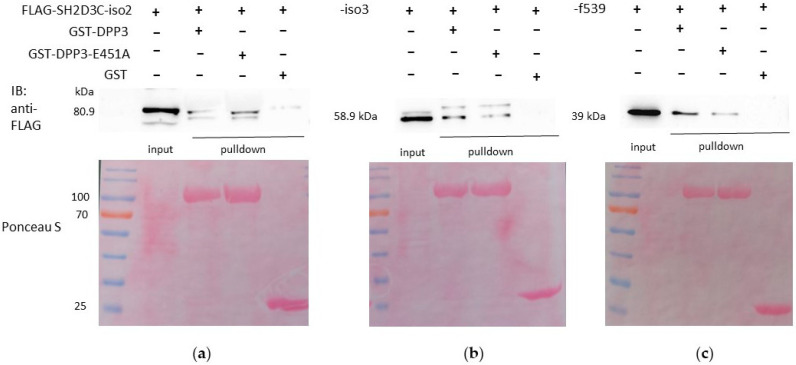
Western blot analysis of GST pulldowns: (**a**) lysates from HEK293T cells overexpressing FLAG-tagged SH2D3C protein isoform 2 (80.9 kDa) were incubated with glutathione agarose beads with bound GST-DPP3, GST-DPP3-E451A and GST, respectively; pulldowns were analyzed by western blot with anti-FLAG antibody; (**b**) lysates from HEK293T cells overexpressing FLAG-tagged SH2D3C protein isoform 3 (58.9 kDa) were incubated with glutathione agarose beads with bound GST-DPP3, GST-DPP3-E451A and GST, respectively; pulldowns were analyzed by western blot with anti-FLAG antibody; (**c**) lysates from HEK293T cells overexpressing FLAG-tagged SH2D3C C-terminal Ras GEF-like domain (f539) were incubated with glutathione agarose beads with bound GST-DPP3, GST-DPP3-E451A and GST, respectively; pulldowns were analyzed by western blot with anti-FLAG antibody; upper panels: anti-FLAG immunoblots; lower panels: Ponceau S staining of the membranes.

**Figure 3 ijms-24-14178-f003:**
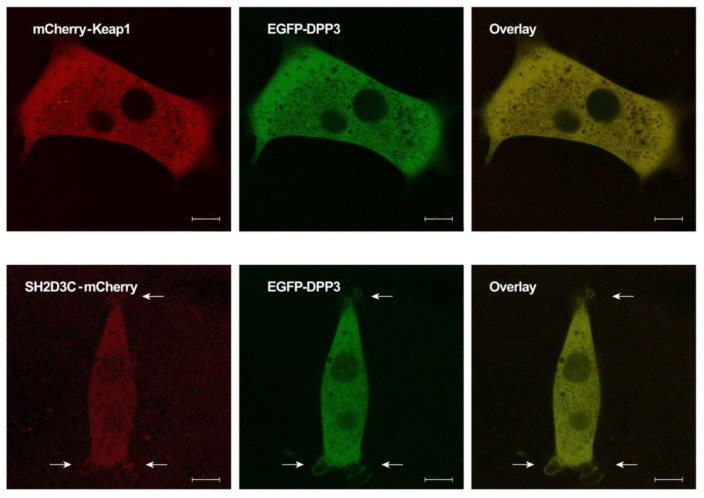
Live-cell imaging of the colocalization (yellow) of mCherry–KEAP1 (red) and EGFP–DPP3 (green) (**upper panel**), and SH2D3C–mCherry (red) and EGFP–DPP3 (green) (**lower panel**) in NIH 3T3 cells; arrows point at membrane ruffles; scale bar 8 µm.

**Figure 4 ijms-24-14178-f004:**
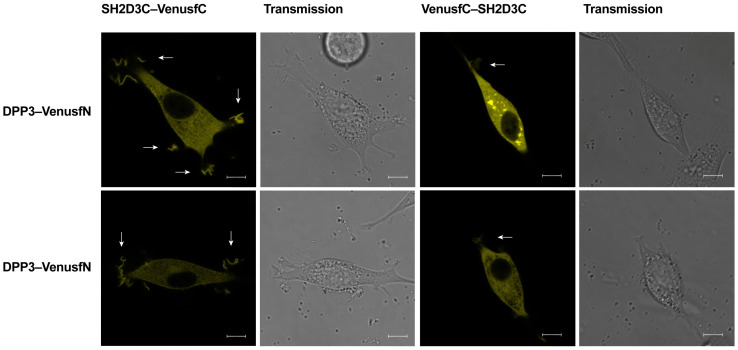
Live-cell imaging of NIH 3T3 cells transiently transfected with vectors expressing DPP3–VenusfN chimera, and SH2D3C–VenusfC (**left panel**) or VenusfC–SH2D3C chimera (**right panel**), respectively. Arrows point at membrane ruffles; scale bar 8 µm.

**Figure 5 ijms-24-14178-f005:**
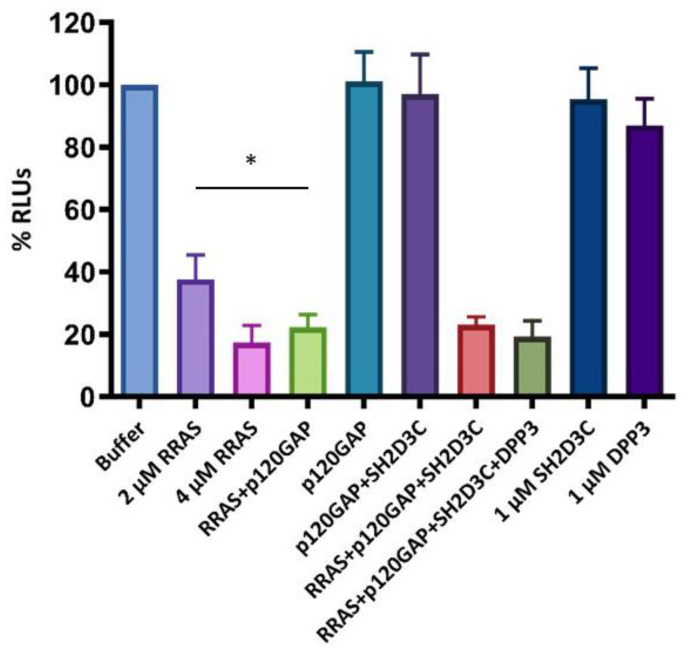
GEF activity test; Histogram showing relative luminescence output compared to buffer with 5 µM GTP control reaction (Buffer). Luminescence output (RLUs) is inversely proportional to the GTPase activity in the reactions. The concentration of proteins, if not stated otherwise, were 2 µM RRAS, 1 µM p120GAP, 1 µM SH2D3C, and 1 µM DPP3. The addition of p120GAP lowers relative luminescence compared to the reaction with only 2 µM RRAS with statistical significance (* *p* = 0.037) confirming that p120GAP acts as RRAS GAP. There was no difference in the relative luminescence between reactions containing only RRAS and p120GAP and reactions that contained putative RRAS GEF, SH2D3C, or SH2D3C and DPP3 in addition to RRAS and p120GAP, respectively. Reactions containing only DPP3 show weak GTPase activity, indicating contamination with bacterial GTPases. Data was analyzed by GraphPad Prism 10 software, and it represents the mean from 3 independent experiments with SD. An unpaired, two-tailed *t*-test was used for statistical analysis.

**Figure 6 ijms-24-14178-f006:**
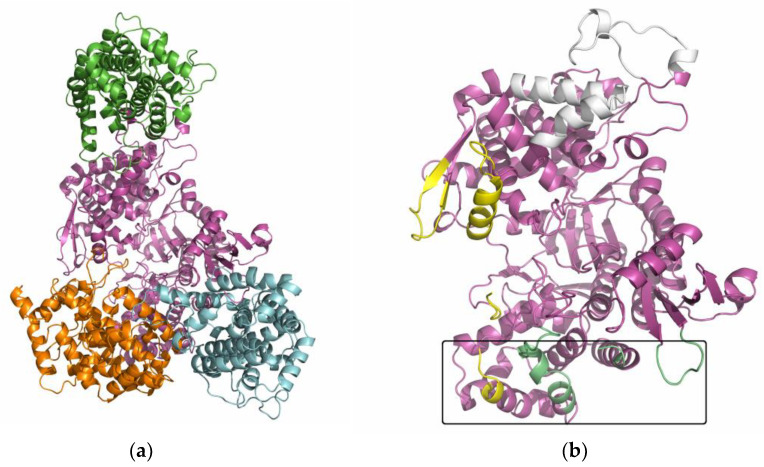
(**a**) Representative poses of the most common alignments of SH2D3C (green, orange and cyan) with respect to DPP3 (magenta) determined by docking; (**b**) DPP3 regions to which SH2D3C binds, “framed” region (the α-helix 139–148, a part of the loop 149–155, and the protein C-terminal) corresponds to the SH2D3C binding site in model 1. The yellow-colored regions (the loop between the two beta strands in the upper domain, residues 587–598, the α-helix 647–663, the unstructured region 115–119, and the protein N-terminal in the lower domain) correspond to the DPP3 region interacting with SH2D3C in models 2–4. The third binding site is located near the ETGE loop (462–484) and two α-helices at the top of the upper domain (613–640), white-colored, and corresponds to the SH2D3C binding site in model 5.

**Figure 7 ijms-24-14178-f007:**
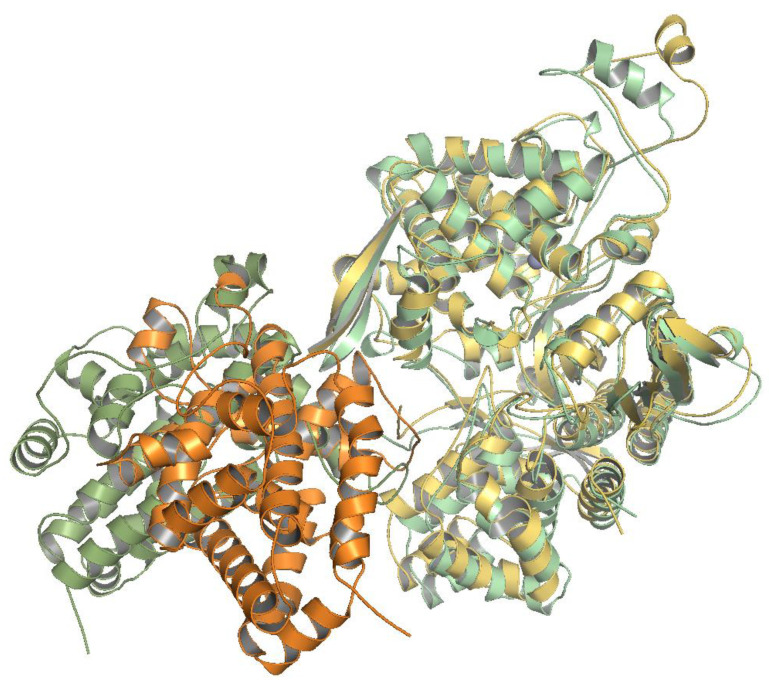
Superposition of the lowest energy structure of model 4 (structure after 450 ns of simulations) and the structure at the end of 700 ns of MD simulations. In the lowest energy structure, DPP3 is colored yellow and SH2D3C is colored orange; in the final structure, both DPP3 and SH2D3C are colored green (light green and smudge).

**Figure 8 ijms-24-14178-f008:**
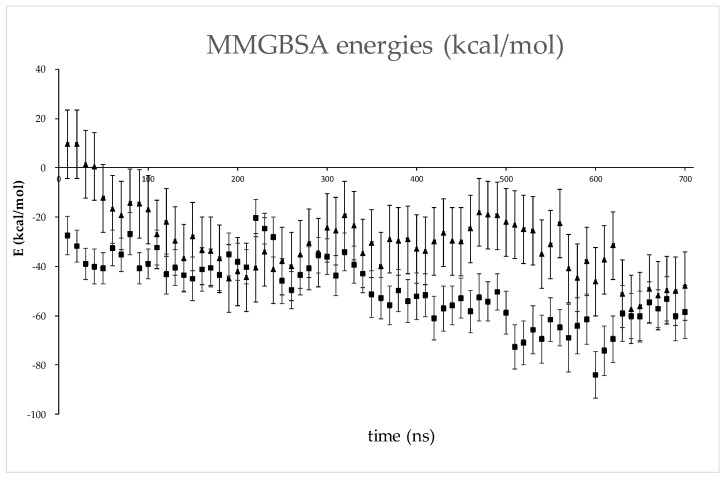
MMGBSA energies (and their standard deviation) for models 3 (▲) and 5 (■). Values were calculated for conformers sampled at 10 ns intervals throughout MD simulations.

**Figure 9 ijms-24-14178-f009:**
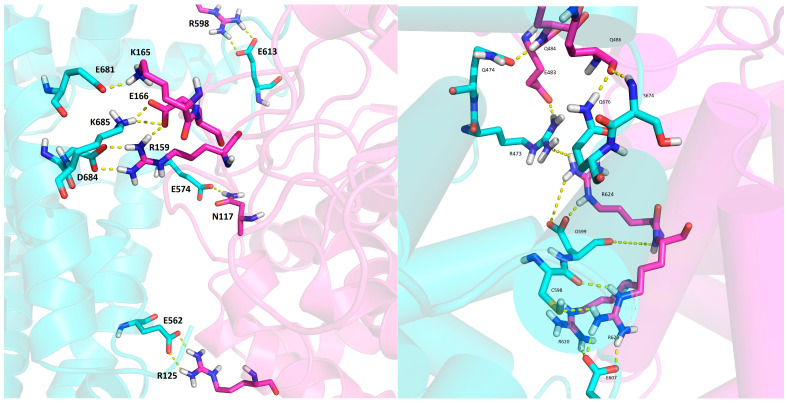
Some of the most populated protein–protein intermolecular interactions in models 3 (**left**) and 5 (**right**). Shown are structures from the region of lowest MMGBSA energies. DPP3 is colored magenta and SH2D3C cyan. The intermolecular hydrogen bonds are shown as dashed yellow lines, and the amino acid residues connected with these hydrogen bonds are shown as sticks, while the rest of the protein is shown as a cartoon.

**Figure 10 ijms-24-14178-f010:**
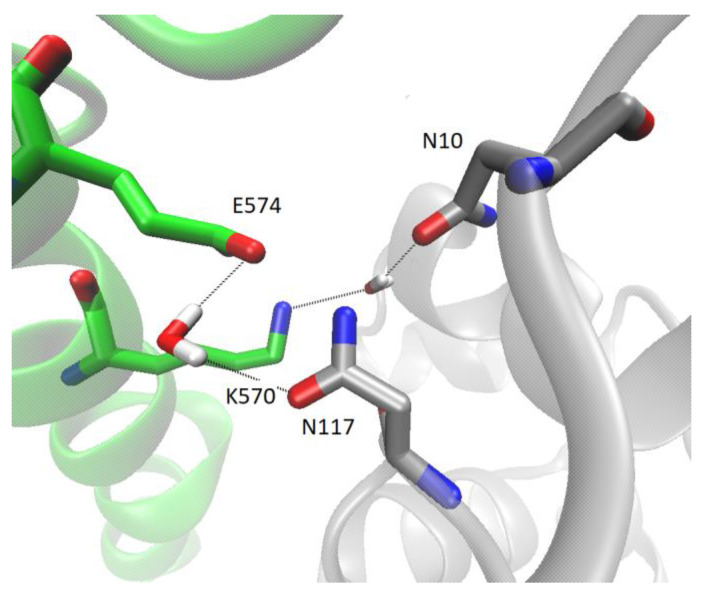
An example of the water-mediated intermolecular hydrogen bonds in model 3. DPP3 is colored gray and SH2D3C green. The hydrogen bonds are shown with thin black lines, and the amino acid residues and water molecules involved in the formation of the hydrogen bond network are shown as sticks, while the rest of the proteins are shown as cartoons.

## Data Availability

Data available on request.

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
