# Peer review of "Identification of SH2 Domain-Containing Protein 3C as a Novel, Putative Interactor of Dipeptidyl Peptidase 3"

_ijms, 2023, doi:10.3390/ijms241814178_

Round 1
Reviewer 1 Report
The manuscript titled "Identification of SH2 domain-containing protein 3C as a novel, 2 putative interactor of dipeptidyl peptidase 3" presents a well-structured and scientifically sound study investigating the interaction between DPP3 and SH2D3C proteins. The study combines experimental and computational approaches, and overall, it contributes valuable insights into this protein-protein interaction. The manuscript is generally well-written and organized, with clear descriptions of the methods employed. However, there are some areas that require further clarification, and I have identified a few points that need attention.
Comments:
· In Figure 4, it would be helpful to label the lanes in the western blot images to specify the bands corresponding to HA-DPP3 and FLAG-SH2D3C isoforms.
· In Figure 5, the y-axis label in the top right panel should specify the unit of measurement for luminescence (e.g., RLUs).
· Specify incubation conditions, such as temperature and duration, for various experimental procedures.
· Include software versions and parameters used for computational methods.
· Clarify the potential physiological relevance of the DPP3-SH2D3C interaction.
· Expand the discussion of computational results, particularly regarding the structural aspects of the interaction.
· There are occasional minor grammatical and typographical errors that should be addressed. For example, there are some missing articles ("the," "an," "a") and prepositions ("in," "on") in certain sentences. These errors, though minor, can affect the overall readability.
The quality of English language in the manuscript is generally good; however, there are some areas where improvements could be made for greater clarity and readability.
Author Response
In Figure 4, it would be helpful to label the lanes in the western blot images to specify the bands corresponding to HA-DPP3 and FLAG-SH2D3C isoforms.
Response:
I assume you meant Figure 1. We did not do anti-HA western blot. We have done a lot of anti-HA co-IPs for HA-DPP3 before this one and we stopped doing anti-HA immunoblots at some point because HA-DPP3 expression was so high that we could usually see HA-DPP3 on the Ponceau stained membrane. Unfortunately, we don't have the photograph of Ponceau stain of this particular membrane, but we are positive that HA-DPP3 is eluted from the beads after co-IP.
- In Figure 5, the y-axis label in the top right panel should specify the unit of measurement for luminescence (e.g., RLUs).
Response:
We have added RLUs to the Figure 5.
- Specify incubation conditions, such as temperature and duration, for various experimental procedures.
Response:
We could not find any procedure for which we did not specify reaction conditions.
4.1.3. Co-immunoprecipitation
Cells were harvested 24-48 hours post transfection in lysis buffer (50 mM Tris, pH 7.5, 75 mM NaCl, 1 mM EDTA, 0.1% Triton X-100) with 1X HALT protease inhibitor cocktail (Thermo Scientific). To each reaction and negative control 20 µl of mouse monoclonal an-ti-HA−agarose beads (Sigma #A2095) was added and incubated at 4 °C for 2-3 hours.
4.1.4. GST-pulldown
GST-DPP3, GST-DPP3-E451A (catalytically inactive mutant) and GST were expressed in BL21-CodonPlus(DE3)-RIL E. coli strain. Bacterial cells were lysed in PBS/1% Tri-ton-X-100 and bound to glutathione-agarose beads (Thermo Scientific) at 4 °C overnight and then washed 3 times with PBS/1% Triton-X-100.
GST-pulldown reactions were incubated for 1 hour at room temperature, washed with PBS/1% NP-40 and the proteins were eluted from the beads with the addition of 20 µl of 1X TB and heating to 98 °C for 5 minutes.
4.1.7. Guanine nucleotide exchange (GEF) activity test
Reactions were incubated for 2 hours at 24 °C, then GTP that remained in the solution was converted to ATP by nucleoside-diphosphate kinase, and generated ATP was used in a luciferase reaction to produce light. Luminescence was measured on multimode plate reader Spark (Tecan).
- Include software versions and parameters used for computational methods.
Response:
The software versions and parameters used for computational methods :
Hybrid bonded/non-bonded parameters for the Zn ion used in this study were previously derived by QM /MM calculations in combination with exhausted MD simulations performed for several metalloenzymes whose metal binding site is similar to that in DPP III. The procedure is described in the reference. (24)
Optimization was performed with the program sander, and heating, density equilibration, and productive MD simulations were performed with pmemd.cuda. Both programs are available in the AMBER18 suit of programs.
- Clarify the potential physiological relevance of the DPP3-SH2D3C interaction.
Response:
We do not know what could be the physiological relevance of this interaction or if it was even physiological. The experiments aimed at elucidation of the physiological role are on the way, however, SH2D3C protein is not expressed in commonly used cell lines like HEK293T, HeLa, RPE-1 and even in cell lines were it has expression (SHSY5Y, Jurkat), the expression is very low and we can barely detect it with the antibodies that are currently available. Studies on SH2D3C published so far were mainly investigation homologue and there are very few papers about human SH2D3C, especially in the last 5 years. Since it is a protein that has role in the regulation of migration and adhesion, we thought it is valuable to publish our findings.
- Expand the discussion of computational results, particularly regarding the structural aspects of the interaction.
Response:
The discussion of computational results is expanded
- There are occasional minor grammatical and typographical errors that should be addressed. For example, there are some missing articles ("the," "an," "a") and prepositions ("in," "on") in certain sentences. These errors, though minor, can affect the overall readability.
Response:
Thank you for mentioning that. We have used Insta-text application to check the English language.

Reviewer 2 Report
The manuscript entitled “Identification of SH2 domain-containing protein 3C as a novel, putative interactor of dipeptidyl peptidase 3”. Certainly, here are some about DPP3 and SH2D3C and their potential interactions.
· What is the known function of DPP3 in the cell, and how does it relate to its role in the final stages of protein turnover?
· Could you explain the significance of DPP3's interaction with KEAP1 and its impact on NRF2 activity and the oxidative stress response?
· How was the role of DPP3 in oxidative stress confirmed in DPP3 knockout (KO) mice, and what were the observed effects on bone health and other physiological aspects?
· Can you elaborate on the increased levels of circulating DPP3 in septic and cardiogenic shock patients and the potential implications for clinical research?
· What prompted the search for novel interactors of DPP3, and how was the SILAC-MS approach employed to identify SH2D3C as a potential interactor?
· Could you provide more information on SH2D3C's functions, including its interactions with EphB2, R-Ras, Rap1A, p130Cas, and other proteins?
· How does the interaction between DPP3 and SH2D3C relate to the regulation of oxidative stress and the cellular processes associated with SH2D3C?
· What are the implications of this newfound interaction between DPP3 and SH2D3C in terms of potential roles in cell adhesion, migration, and growth, as well as in the KEAP1-NRF2 pathway?
· What are the experimental methods employed to investigate the interaction between DPP3 and SH2D3C isoforms 2 and 3?
· Can you explain the significance of the GST-pull-down assay and how it confirmed the interaction between DPP3 and SH2D3C?
· What is the subcellular localization of EGFP-DPP3 and SH2D3C-mCherry, and how was it visualized in NIH3T3 cells?
· How does the bimolecular fluorescence complementation (BiFC) assay provide evidence for the direct interaction between DPP3 and SH2D3C?
· What is the rationale behind testing the GEF activity of SH2D3C towards small GTPase RRAS, and what were the results of this assay?
· Could you elaborate on the different models of binding between DPP3 and the C-terminal domain of SH2D3C revealed by protein docking analysis?
· What implications do these different binding models have for the functional interaction between DPP3 and SH2D3C?
· What criteria were used to select the complexes for molecular dynamics (MD) simulations, and why was the mutual orientation of SH2D3C and p130cas in the crystallographically determined structure considered?
· What were the key findings or observations during the MD simulations that led to the selection of certain models for further investigation?
· Can you describe the changes in the orientation of SH2D3C during the last 140 ns of the MD simulations in model 4 and its impact on the binding free energy of DPP3?
· How does the radius of gyration (Rgyr) provide insights into the structural dynamics of the protein complexes, and what do the Rgyr values indicate for models 3, 4, and 5?
· What is the significance of the binding free energies calculated using the LIE (Linear Interaction Energy) method, and how does it compare to the MMGBSA energies in selecting the most stable model?
· Can you explain the differences in solvation components of binding between model 3 and model 5 and how they affect the overall binding free energies?
· What were the most populated polar intermolecular hydrogen bonds identified between DPP3 and SH2D3C in models 3 and 5, and how do they contribute to the stability of the complexes?
· Could you provide more information on the water-mediated intermolecular hydrogen bonds observed in model 3 and their significance in protein-protein interactions?
· What insights or implications do the protein-protein interactions and structural dynamics observed in models 3 and 5 have for the functional interaction between DPP3 and SH2D3C?
· What is the significance of testing SH2D3C for GEF activity, and how does the presence of DPP3 affect this activity?
· Can you explain the experimental setup used to assess SH2D3C GEF activity toward RRAS, including the components of the reaction mix and the rationale behind their inclusion?
· What were the specific results obtained from the GTPase-Glo Assay, and how were they interpreted to assess GEF activity?
· Were there any significant differences in GEF activity when comparing SH2D3C alone, SH2D3C with DPP3, and control reactions? If so, what were those differences?
· What motivated the selection of the experimentally determined structures of SH2D3C and the human DPP3 for use in molecular docking experiments?
· What are the key differences or unique characteristics of the crystallographic 'open' structure of DPP3 (PDB_ID: 3FVY) compared to the semi-open structure used in the molecular simulations?
· How many molecular docking servers were used in the study, and what were the criteria for selecting these servers?
· Could you explain the significance of adjusting the protonation states of charged residues and histidines to a pH of about 7.5 in the molecular modeling, and how this choice reflects physiological conditions?
· What specific criteria or considerations were used to determine the protonation states of histidines, especially those with both Nδ and Nε protonated?
· Can you elaborate on the role of the zinc ion in the active site of DPP3 and how it was described using the hybrid bonded/non-bonded model for metallopeptidases?
· How was the solvation of the system carried out, and why was a truncated octahedron of OPC-water molecules used for this purpose?
· What was the rationale behind adding Na+ (and/or Cl-) ions to achieve electroneutrality or the desired salt concentration in the system, and how was this accomplished?
· In the optimization cycles prior to productive MD simulations, why were different restraints applied in each cycle (e.g., constraints on protein and zinc ion, constraints on protein backbone, no additional restraints), and how does this affect system relaxation? What is software utilized?
· The author should update the methodology reference, which can be seen PMID: 30602415, 29765814, 33131721, 34551668, 37071766.
· Could you provide more details about the heating process, including the NVT ensemble used and any specific conditions applied to reach 300 K?
· What was the motivation for using a time step of 0.5 fs during heating simulations and 1 fs during equilibration and production MD simulations?
· What specific parameters and criteria were used to calculate geometry parameters (RMSD, Rgyration, RMSF), perform hydrogen bonding analysis, and analyze LIE (Ligand Interaction Energy) throughout the MD simulations?
· In RMSD half black trajectory, why?
· What was the purpose of conducting 2 μs of productive classical MD simulations for the predicted DPP III - SH2D3C complexes?
· How did you determine the dielectric constant of 2 and the ionic strength of 0.100 mol dm-3 used in the MMGBSA energy calculations?
Moderate improvement required!
Author Response
What is the known function of DPP3 in the cell, and how does it relate to its role in the final stages of protein turnover?
Response:
It is assumed that DPP3 has a function in the final stages of protein turnover because it cleaves 4 to 8 amino acid long peptides with broad specificity and it is ubiquitously present in almost every cell in human (it is part of central human proteome) and in organism from bacteria to human, however, it would be very hard to confirm that experimentally, since it is not the only dipeptidyl peptidase and it is not essential for the cells (KO mice are viable). Other potential functions of DPP3 are regulation of blood pressure through the digestion of angiotensins (ANGII, ANG1-7 and ANG1-5), regulation of pain (digestion of endomorphins and enkephalines) and regulation of oxidative stress response through NRF2-KEAP1 signaling pathway. None of those roles has been completely proved in vivo, except the involvement in the oxidative stress response mediated through NRF2 in osteoclasts (DPP3 KO mice have a disorder in bone development which was related to increase of ROS in osteoclasts and decreased activity of NRF2). Pathophysiological roles of DPP3 are very nicely reviewed in Malovan, G.; Hierzberger, B.; Suraci, S.; Schaefer, M.; Santos, K.; Jha, S.; Macheroux, P. The emerging role of dipeptidyl peptidase 3 in pathophysiology. FEBS J. 2022, 1–17, doi:10.1111/febs.16429.
- Could you explain the significance of DPP3's interaction with KEAP1 and its impact on NRF2 activity and the oxidative stress response?
Response:
DPP3 is one of the proteins that interacts with KEAP1 through its ETGE motif and causes the upregulation of NRF2 activity. It was identified as one of 2 proteins (the other one being p62) with the strongest impact on NRF2-ARE transcription back in 2007 (Liu, Y.; Kern, J.T.; Walker, J.R.; Johnson, J.A.; Schultz, P.G.; Luesch, H. A genomic screen for activators of the antioxidant response element. Proc. Natl. Acad. Sci. U. S. A. 2007, 104, 5205–5210, doi:10.1073/pnas.0700898104), however, the mechanisms of its involvement were clarified in 2013 by Hast et al (1. Hast, B.E.; Goldfarb, D.; Mulvaney, K.M.; Hast, M.A.; Siesser, P.F.; Yan, F.; Hayes, D.N.; Major, M.B. Proteomic analysis of ubiquitin ligase KEAP1 reveals associated proteins that inhibit NRF2 ubiquitination. Cancer Res. 2013, 73, 2199–2210, doi:10.1158/0008-5472.CAN-12-4400), who showed that DPP3 interacts with KEAP1 through ETGE and inhibits NRF2 ubiquitination. Lu et al. confirmed this finding in 2017 and discovered that DPP3-KEAP1 interaction is induced by H2O2 treatment of cells and that higher expression of DPP3 in ER+ breast cancer is related to higher NRF2 activity and poor prognosis (Lu, K.; Alcivar, A.L.; Ma, J.; Foo, T.K.; Zywea, S.; Huo, Y.; Kensler, T.W.; Gatza, M.L.; Xia, B. DPP3 in NRF2 Signaling and Breast Cancer. Free Radic. Biol. Med. 2017, 100, S132, doi:10.1016/j.freeradbiomed.2016.10.348).
- How was the role of DPP3 in oxidative stress confirmed in DPP3 knockout (KO) mice, and what were the observed effects on bone health and other physiological aspects?
Response:
Lack of DPP3 caused increased levels of ROS and decreased levels of NRF2 and its target HO-1. You can find the details in the Menale, C.; Robinson, L.J.; Palagano, E.; Rigoni, R.; Erreni, M.; Almarza, A.J.; Strina, D.; Mantero, S.; Lizier, M.; Forlino, A.; et al. Absence of Dipeptidyl Peptidase 3 Increases Oxidative Stress and Causes Bone Loss. J. Bone Miner. Res. 2019, 34, 2133–2148, doi:10.1002/jbmr.3829.
- Can you elaborate on the increased levels of circulating DPP3 in septic and cardiogenic shock patients and the potential implications for clinical research?
Response:
DPP3 was first detected in blood circulation in 2019 when the specific test for its activity and concentration in blood plasma and serum were developed. Since than a number of papers was published in which higher concentration of DPP3 were related to adverse outcomes of several acute conditions, including cardiac and septic shock, kidney failure etc. It was also shown in animal models that treatment with DPP3 reduces cardiac and renal function, while treatment with anti-DPP3 antibody improves cardiac function and reduces mortality. You can also find more information about this in Malovan et al. 2022 review.
- What prompted the search for novel interactors of DPP3, and how was the SILAC-MS approach employed to identify SH2D3C as a potential interactor?
Response:
KEAP1 is the only interactor confirmed by several groups, so far, and back in 2015 when we started the study it was the only confirmed interactor. We thought that there has to be more protein interactors of DPP3 and that the discovery of novel interactors might give us clues for the elucidation of its physiological role(s). SILAC-MS was performed by our former collaborators in Frankfurt, Germany. They developed TRex HEK293T cell lines stably expressing HA-DPP3 and also TRex HEK293T cell line stably transfected with corresponding empty vector. They grew HA-DPP3-overexpressing cells on medium with heavy Arg and Lys and cells stably transformed with empty vector on normal medium. They lysed the cells, mixed lysates and did anti-HA co-IP to precipitate HA-DPP3 and all proteins bound to it. Than they did MS analysis and determined the intensity signal for peptides from DPP-ox cells and empty vector controls and calculated the ratio of intensities in DPP3 vs EV cells for each protein. They made 4 biological replicates of the experiment. We don’t know the exact details of how they did the calculation, we were just provided with Excel files with normalized ratios and with a list of protein with DPP3 vs EV ratio > 1.5. Since none of the proteins identified as hits had any previous connection with DPP3 (except KEAP1 which was found in one replicate), we analyzed the proteins found in 3 replicates (only ubiquitin was found in all 4) by co-IP and Y2H and when we didn’t confirm any of the interactions, we analyzed 9 more proteins found in 2 replicates, but with highest DPP3 vs EV ratio, however, we couldn’t confirm any of the interactions with co-IP. Than we decided to analyze SH2D3C that was found in only one replicate and we confirmed the interactions with several methods.
- Could you provide more information on SH2D3C's functions, including its interactions with EphB2, R-Ras, Rap1A, p130Cas, and other proteins?
Response:
Most of the papers regarding SH2D3C function were investigating mouse protein, and most of them were published from 15 to almost 25 years ago. Dodelet et al. 1999 discovered new SH2-containing mouse protein, named it SHEP1 (alt. name of SH2D3C) and showed that it binds phosphorylated EphB2 receptor through its SH2 domain and that co-expression of both proteins leads to tyrosine phosphorylation of SHEP1. They also showed that C-terminal Ras GEF domain of SHEP1 binds small GTPases Rap1a and RRas, but shows no GEF activity. One of their assumptions was that maybe SHEP1 acts as competitive inhibitor of Ras GEF proteins that activate Rap1a and RRas, however, that was never proved. Sakakibara et al. 2000 identified Chat protein (another alt. name of SH2D3C) in cell-cell adherence junction and demonstrated that binds 2 proteins from Cas family, Cas and HEF1. Chat was phosphorylated by MAP kinase after the stimulation of tyrosine receptors in PC12 cells. They detected the expression of Chat in several mouse tissues and identified a longer variant, named Chat-H (corresponding to human isoform 1) that is expressed in hematopoietic tissue. They also showed that Chat and Cas are located mainly in cytoplasm in Cos7 cells, but they translocate to membrane ruffles when cells are stimulated with EGF. Overexpression of both Cas and Chat, respectively, activated c-Jun N-terminal kinase (JNK). They concluded that Chat integrates signals from tyrosine kinases and MAP kinases and controls growth and cytoskeletal organization through Cas pathway. Chat-H was also shown to regulate chemokine induced T-cell migration and adhesion (Regelman et al. 2006). One of the most recent studies showed that Chat-H stimulates the migration of monocytes to atherosclerotic plaques and formation of plaques (Herbin, O.; Regelmann, A.G.; Ramkhelawon, B.; Weinstein, E.G.; Moore, K.J.; Alexandropoulos, K.; Weinstein, E.G.; Regelmann, A.G.; Herbin, O.; Regelmann, A.G.; et al. Monocyte adhesion and plaque recruitment during atherosclerosis development is regulated by the adapter protein chat-H/SHEP1. Arterioscler. Thromb. Vasc. Biol. 2016, 36, 1791–1801, doi:10.1161/ATVBAHA.116.308014.). The investigation of human SH2D3C confirmed the binding to Cas family proteins and revealed that NSP family proteins (SH2D3A/NSP1n BCAR3/SH2D3B/NSP2 and SH2D3C/NSP3/Chat/SHEP1) bind Cas family of proteins (p130Cas, HEF1/NEDD9, EFS) in a promiscuous manner and form multidomain signaling complexes. It was also shown that Ras GEF like domain of BCAR3 and SH2D3C adopts closed structure which makes it inactive, but stimulates the binding to Cas family proteins (Mace et al 2011). One study of SH2D3C KO mice found that its olfactory sensory axons do not penetrate the forebrain which causes the disruption of its connection with olfactory bulb so it does not develop properly (Wang, L.; Vervoort, V.; Wallez, Y.; Coré, N.; Cremer, H.; Pasquale, E.B. The Src homology 2 domain protein shep1 plays an important role in the penetration of olfactory sensory axons into the forebrain. J. Neurosci. 2010, 30, 13201–13210, doi:10.1523/JNEUROSCI.3289-10.2010.). Another investigation of SH2D3C KO mice showed that the defects are present in marginal zone B cells function, caused by their reduced mobility which might be the consequence of the distorted integrin signalling, but T cell development and function was not influenced by the lack of SH2D3C (Al-Shami, A.; Wilkins, C.; Crisostomo, J.; Seshasayee, D.; Martin, F.; Xu, N.; Suwanichkul, A.; Anderson, S.J.; Oravecz, T. The Adaptor Protein Sh2d3c Is Critical for Marginal Zone B Cell Development and Function. J. Immunol. 2010, 185, 327–334, doi:10.4049/jimmunol.1000096.), which is in contradiction with some other research that found it indispensable for cell adhesion and T cell migration (Regelmann et al. 2006). Another confirmation that SH2D3C is important for marginal zone B cells development came from the study of Browne et al, 2010. They showed that SH2D3C deficient B cells do not migrate to the marginal zone which prevents their normal development. Those cells are also unresponsive to the chemokine and Sphingosine-1-phosphate (S1P) treatment, probably due to the reduced CasL phosphorylation (Browne et al. 2010). SH2D3C is positively regulated by Rx1 transcription factor in the retinal precursor cells, together with EphrinB1, and it is assumed that it has a role in the regulation of migrations of retinal progenitors and/or evagination of optic vesical (Giudetti et al. 2014). Several recent studies found that SH2D3C deletion is associated with longevity in Chinese population (Zhao et al. 2018), and that it was upregulated in rat cortical neurons culture upon treatment with amyloid-ẞ oligomers, while higher protein levels of SH2D3C were also detected in Alzheimer disease model mice compared to the wild type. The overexpression of SH2D3C also induced neuronal death (Gomi, Uchida, and Endo 2018). In summary, there are a lot of findings regarding SH2D3C’s involvement in cell migration, adhesion, tissue organization, and the regulation of the immune response, however, the exact mechanisms of its involvement in these processes are not completely clarified.
- How does the interaction between DPP3 and SH2D3C relate to the regulation of oxidative stress and the cellular processes associated with SH2D3C?
Response:
We don’t have the answer to that question, yet. This is under investigation at the moment, however the investigations are hindered by the lack of good cell models, since we cannot detect SH2D3C protein expression in a number of cell lines that we tested, including HEK293T, HeLa, RPE-1, several cancer cell lines from different types of cancer, including breast, colorectal, cervical etc. We detected the SH2D3C protein in Jurkat and SHSY5Y cells, however, its expression is very low. We are aware that the study is not finished, but we think that the results obtained so far speak strongly in favor of the interaction and might be interesting to the people in the field of DPP3, which is broadening lately.
- What are the implications of this newfound interaction between DPP3 and SH2D3C in terms of potential roles in cell adhesion, migration, and growth, as well as in the KEAP1-NRF2 pathway?
Response:
As we have wrote in the response to previous comment, we don’t know, yet.
- What are the experimental methods employed to investigate the interaction between DPP3 and SH2D3C isoforms 2 and 3?
Response:
Interaction was investigated by co-IP, GST-pulldown and BiFC.
- Can you explain the significance of the GST-pull-down assay and how it confirmed the interaction between DPP3 and SH2D3C?
Response:
We expressed DPP3 as a chimeric protein with GST on the amino-end in E. coli. We also expressed catalytically-inactive form of DPP3, E451A with GST and GST alone as negative control. Proteins (GST-DPP3, GST-DPP3-E451A and GST, respectively) were bound to glutathione-agarose beads (GST binds glutathione) and unbound proteins were washed. Beads with bound proteins were mixed with cell lysates from HEK293T cells overexpressing FLAG-tagged isoforms of SH2D3C, isoforms 2 and 3 (we didn’t express the longest isoform 1 because its expression in HEK293T cell was low) and we also expressed only the C-terminal domain of SH2D3C, respectively. Each lysate from HEK293T cells was mixed with 3 different proteins bound to beads, respectively. After incubation and wash, bound proteins were eluted from the beads, and western blot with anti-FLAG antibody was used to check if SH2D3C has bind to DPP3 or DPP3-E451A. GST served as negative control and neither protein should bind to it. Western blot analysis showed that both isoform 2 and 3 and C-terminal domain of SH2D3C bind to both DPP3 and DPP3-E451A and none of the proteins bind to GST. Therefore, we confirmed the binding of SH2D3C to DPP3.
- What is the subcellular localization of EGFP-DPP3 and SH2D3C-mCherry, and how was it visualized in NIH3T3 cells?
Response:
EGFP-DPP3 and SH2D3C-mCherry were overexpressed in NIH 3T3 cells and the signals from EGFP and mCherry, and their overlay were analyzed by confocal microscopy. Signals were detected predominantly in the cytosol, however, weaker staining can be seen in the nucleus and there are also signals from both proteins was also detected in membrane ruffles, structures involved in cell migration.
- How does the bimolecular fluorescence complementation (BiFC) assay provide evidence for the direct interaction between DPP3 and SH2D3C?
Response:
You are right, we cannot claim that the interaction is direct based on the results of BiFC assay. It is possible that DPP3 and SH2D3C are part of the protein complex that brings them in sufficiently close proximity to allow for the 2 fragment of VenusN to merge. We have revised the sentence in the manuscript.
Old version:
To investigate whether SH2D3C and DPP3 interact directly we employed a bimolecular fluorescence complementation (BiFC) assay.
New version:
To investigate whether SH2D3C and DPP3 interact specifically in the live cells, we employed a bimolecular fluorescence complementation (BiFC) assay.
- What is the rationale behind testing the GEF activity of SH2D3C towards small GTPase RRAS, and what were the results of this assay?
Response:
SH2D3C contains C-terminal Ras GEF-like domain which is inactive, although it was found that mouse protein binds small Ras GTPases, Rap1a and RRas. Mace et al. 2011 obtained the crystal structure of Ras GEF-like domain of SH2D3C bound to C-terminal domain of Cas (p130Cas, BCAR1) protein and found that Ras GEF-like domain of SH2D3C adopts a closed structure (compared to active Ras GEF domains) which renders it inactive. They analyzed Ras GEF activity of SH2D3C and found that it is inactive, but they also analyzed the activity with Cas protein because they thought that it could activate SH2D3C, howeverm they it didn’t. Our idea was similar, we thought that maybe the interaction with DPP3 might activate SH2D3C Ras GEF activity, however, we also did not show that, however, we had to use low concentrations of DPP3, because we got the GTPase activity even with only DPP3 in the reaction. We assume that DPP3 purified from E. coli is contaminated with bacterial GTPases.
- Could you elaborate on the different models of binding between DPP3 and the C-terminal domain of SH2D3C revealed by protein docking analysis?
Response:
At least 10 conformations were generated with each of the servers mentioned in ‘Material and Methods’ section, ClusPro, Haddock and GRAMM-X. The best docking results were selected according to their docking score, and the results obtained by the different servers were compared and visualized. The overlap of the best rated protein-protein docking results is shown in Fig. Supp-dock. ClusPro mainly predicted the binding of SH2D3C to the lower domain of DPP3, sometimes also to the upper domain of DPP3 near the β-strands (yellow colored regions in Fig. 5). Haddock and GRAMM-X results also predicted binding of SH2D3C to the yellow-stained regions of DPP3, but also to the regin near the ETGE loop of DPP3 (colored white in Fig. 5).
- What implications do these different binding models have for the functional interaction between DPP3 and SH2D3C?
Response:
It is difficult to predict what effects different binding models of DPP3 – SH2D3C would have on their functional interactions because the full structure of SF2D3C is not available, only its Ras GEF-like domain. However, we could assume that if SH2D3C binds to DPP3 at the regions colored yellow in Figure 5 (as in models 1-4), their interaction would not significantly affect the interaction of DPP3 with KEAP1 protein, because in these models DPP3 interacts with SH2D3C via regions distant from the region with which it interacts with KEAP1, so in this case DPP3 could interact with both proteins simultaneously. However, binding of SH2D3C to a region near the DPP3 ETGE loop would affect the interaction between DPP3 and KEAP1.
- What criteria were used to select the complexes for molecular dynamics (MD) simulations, and why was the mutual orientation of SH2D3C and p130cas in the crystallographically determined structure considered?
Response:
All non-redundant high-scoring models (i.e., all low-energy models binding either to different regions of DPP3 or in significantly different orientations) obtained by molecular docking were subjected to MD simulations. The SH2D3C - p130cas complex is the only complex of SH2D3C with a protein that was experimentally determined. Although p130cas and DPP3 are not structurally or functionally related, it is reasonable to assume that the region of SH2D3C to which p130cas binds is also suitable for binding of the other proteins. Moreover, both p130 and DPP3 bind to this region with alpha helices.
- What were the key findings or observations during the MD simulations that led to the selection of certain models for further investigation?
Response:
Stability of the simulated complexes, quantified by the calculations of the binding affinity by MMGBSA method, was used as main criteria for the selection of complexes for further analysis
- Can you describe the changes in the orientation of SH2D3C during the last 140 ns of the MD simulations in model 4 and its impact on the binding free energy of DPP3?
Response:
During the last 140 ns of the MD -simulations of model 4, the mutual orientation of DPP3 and SH2D3C has changed significantly, leading to an increase in binding free energy. After 560 ns of the MD simulations, SH2D3C moved away from the lower domain of DPP3, and the strength of their interaction decreased, as can be seen from the values of the LIE energies in Table S4. The binding free energy calculated by the MMGBSA method also became less favorable. Comparing the MM (ΔGgass) energy (change in protein-protein interaction energy) and the GBSA energy (ΔGsol) (change of free energy of solvation), it can be seen that the former energy increased (from ca. -150 kcal/mol to values > 0) and the other decreased from values around 100 kcal/mol to values around -30 kcal/mol, indicating that after 560 ns of the MD simulations, the protein-protein interface decreased and they became more solvated.
- How does the radius of gyration (Rgyr) provide insights into the structural dynamics of the protein complexes, and what do the Rgyr values indicate for models 3, 4, and 5?
Response:
In our study, Rgyr is defined as the root-mean-square distance of the backbone atoms of the proteins from the center of mass of the complex. Changes in Rgyr indicate either changes of the structure of the interacting protein (one or both) and/or on the changes of the mutual orientation of the proteins. To obtained better insight into the changes (scenario), it is necessary to analyze Rgyr of the complex and of each interacting protein separately. If Rgyr of the interacting proteins have not followed the changes of Rgyr of the complex, this indicates that the mutual orientation of the interacting proteins changed. Also, the significant increase of the Rgyr value of a complex could indicate a separation of the proteins (disruption of the complex). During the MD simulations, Rgyr of SH2D3C did not change significantly (only a slight increase of SH2D3C was detected during the simulations of model 4), so its value does not affect Rgyr of the complex. On the other hand, Rgyr of DPP3 decreased in model 3 during the simulation of MD and increased in models 4 and 5, with a larger change observed in the simulation of model 5. However, Rgyr of model 3 did not change significantly during the simulation, suggesting that the mutual orientation of the proteins did not change significantly, but DPP3 became more globular during the simulation. In contrast, Rgyr increased in models 4 and 5, and because this increase is larger than the DPP3 Rgyr increase, it is reasonable to assume that the orientation of the interacting proteins changed.
- What is the significance of the binding free energies calculated using the LIE (Linear Interaction Energy) method, and how does it compare to the MMGBSA energies in selecting the most stable model?
Response:
The approach LIE assumes that the free energy of binding can be calculated from the difference between the interaction energies of the ligand when it is bound to the protein and that when it is in solution and is given by the following equation
The binding free energy difference calculated using MMGBSA is given by equation:
Where difference of the polar component of solvation is calculated using GB (Generalized Bohr) method and difference of nonpolar component is given by equation:
In contrast to MMGBSA, the LIE approach does not explicitly account for target desolvation, which is partly a consequence of ligand binding. Therefore, the difference in the LIE energies calculated for different complexes of the same partners is actually a difference in their interaction energies. On the other hand, the differences in MMGBSA energies include differences in free solvation energies in addition to differences in interaction energies.
- Can you explain the differences in solvation components of binding between model 3 and model 5 and how they affect the overall binding free energies?
Response: De-solvation of proteins upon binding is much less favorable in the case of model 3 than in model 5 (Fig. S15). The change in solvation energy upon binding of proteins is given as:
For model 3 the average calculated for the period of 200 ns of MD simulation, by MMGBSA method is about 500 kcal/mol, and for model 5 it is about 440 kcal/mol. That is, from the point of view of solvation energy, the binding of SH2D3C to DPP3 is unfavorable, but less unfavorable in the case of model 5 than in the case of model 3.What is the reason for this? The strongest interactions between two proteins are established between their polar amino acid residues (or polar patches on the protein surface). Because the protein surface area desolvated during complex formation is larger in model 3 than in model 5 (see Fig. S15), the change in free energy of solvation upon protein binding is higher (about 60 kcal7mol more positive, less favorable change) than in model 5.In MMGBSA method the polar component of is calculated using GB method and non polar using equation:
∆Gsolv =∆GGB +∆Gsolv-np=∆GGB +ηSASA+β, where is related to η and β are empirically determined parameters
- What were the most populated polar intermolecular hydrogen bonds identified between DPP3 and SH2D3C in models 3 and 5, and how do they contribute to the stability of the complexes?
Response:
The most polar hydrogen bonds in model 3 are between the guanidino group of R598 of DPP3 and the carboxyl group of E613 of SH2D3C and between the guanidino group of R125 of DPP3 and the carboxyl group of E562 of SH2D3C, wherein R598 is on the β-strand of the upper DPP3 domain and R125 on the α-helix from the bottom of the lower DPP3 domain. The most polar hydrogen bond in model 5 is between the guanidino group of R624 at the top of the upper (catalytic) domain of DPP3 and the carboxyl group of E607 from SH2D3C.
- Could you provide more information on the water-mediated intermolecular hydrogen bonds observed in model 3 and their significance in protein-protein interactions?
Response:
Since in model 3 SH2D3C interacts with amino acid residues from both DPP3 domains, the upper and the lower, there is a lot of space in between that is filled by the water molecules that make polar contacts and hydrogen bonds with the amino acid residues of the interacting proteins, and in some cases the same water molecules interact with the amino acid residues of the two proteins and bridge their interaction (forming the bridge between them) (see Figure S-new).
- What insights or implications do the protein-protein interactions and structural dynamics observed in models 3 and 5 have for the functional interaction between DPP3 and SH2D3C?
Response:
The interactions between DPP3 and SH2D3C observed in model 5 and the structural dynamics would significantly affect the interaction of DPP3 with the KEAP1 protein, because in this model DPP3 interacts with SH2D3C via the same region as with the Kelch domain of KEAP1. On the other hand, the region through which DPP3 interacts with SH2D3C in model 3 is far from the region through which it interacts with KEAP1, so DPP3 would be able to interact with both proteins simultaneously in this case. However, this binding mode could affect enzyme activity because it involves regions on both protein domains and, in this way, rigidifies the long-range dynamics of the protein, which are described as movements between domains (closing and opening) and are important for substrate binding and product release.
- What is the significance of testing SH2D3C for GEF activity, and how does the presence of DPP3 affect this activity?
Response:
SH2D3C has a Ras GEF-like domain that is inactive because it adopts closed structure compared to the active GEF domains. We thought that the interaction of SH2D3C with DPP3 might cause the conformational change that is going to open the structure and activate SH2D3C, however, we could not detect SH2D3C GEF activity in the presence of DPP3.
- Can you explain the experimental setup used to assess SH2D3C GEF activity toward RRAS, including the components of the reaction mix and the rationale behind their inclusion?
Response:
We used commercially available kit for GEF activity test, GTPase-Glo Assay (Promega). This is the manufacturer’s brief description of the kit and the procedure:
“The GTPase-Glo™ Assay assesses the activities of GTPases, GAPs and GEFs, which are components of the GTPase cycle, by detecting the amount of GTP remaining after GTP hydrolysis in a GTPase reaction. The remaining GTP is converted to ATP using the GTPase-Glo™ Reagent, and the ATP is then detected using a proprietary thermostable luciferase (Ultra-Glo™ Recombinant Luciferase) and luciferin substrate to produce bioluminescence. The kit contains optimized reaction buffers, GTPase/GAP Buffer and GEF Buffer, for performing GTPase and GAP reactions and GEF reactions, respectively. These two buffers primarily differ in their Mg2+ content, which is critical for nucleotide loading and unloading of the GTPase, thereby affecting the GTPase cycle. With the GTPase-Glo™ Assay, you can measure intrinsic GTPase activity, GAP-stimulated GTPase activity, GAP activity and GEF activity. GTPase, GAP and GEF activity is inversely correlated to the amount of light produced. A highly active GTPase hydrolyzes more GTP, reducing the amount of ATP produced from GTP and reducing light output. A less active GTPase hydrolyzes less GTP, leaving a larger amount of GTP to be converted to ATP and producing more light.”
Since we measured GEF activity, we needed to have all the proteins involved in the GTPase cycle in the mixture (small GTPase Ras, Ras GAP and Ras GEF). We had control reactions with only RRas (2 different concentrations) (to test if our RRas is active), RRas+p120GAP (to test if p120GAP acts as GAP for RRas which has not been shown before), p120GAP+SH2D3C (to check if GTPase activity is not unspecific). Our sample reactions were RRas+p120GAP+SH2D3C and RRas+p120GAP+SH2D3C+DPP3. DPP3 was added in the second reaction mix because we wanted to check if the addition of DPP3 is going to activate SH2D3C’s GEF activity. We also had 2 additional controls, SH2D3C and DPP3 alone, because we detected high GTPase acitivity when we used higher amounts of DPP3 in the reaction (when we used 20 uM DPP3, almost all GTP was converted to GDP), so we had to make sure that the potential GTPase activity is the result of SH2D3C’s GEF activity and not the contamination of DPP3 with GTPases.
- What were the specific results obtained from the GTPase-Glo Assay, and how were they interpreted to assess GEF activity?
Response:
The specific result is the luminescence measured for each reaction. The luminescence is the measure of the residual GTP in the reaction. If there’s no GTPase activity than the luminescence is the same in the reactions as in the buffer control. Lower luminescence corresponds to higher GTPase activity.
- Were there any significant differences in GEF activity when comparing SH2D3C alone, SH2D3C with DPP3, and control reactions? If so, what were those differences?
Response:
The luminescence output in our experiment was the same in the control reaction with RRas+p120GAP as in the reaction with RRas, p120GAP and SH2D3C and RRas, p120GAP, SH2D3C and DPP3, respectively, so we concluded that SH2D3C does not have GEF activity (which was shown before) and that DPP3 does not activate SH2D3C in the in vitro reactions.
- What motivated the selection of the experimentally determined structures of SH2D3C and the human DPP3 for use in molecular docking experiments?
Response:
3T6G is the only structure of SH2D3C that has been experimentally determined. The motivation for selecting the open and semi-open conformation of DPP3 is based on our previous studies, namely we have shown that the semi-open conformation of DPP3 is the most abundant in aqueous solution.
- What are the key differences or unique characteristics of the crystallographic 'open' structure of DPP3 (PDB_ID: 3FVY) compared to the semi-open structure used in the molecular simulations?
Response:
DPP3 consists of two domains, the catalytic domain and the ligand-binding domain. Each of these domains has the same structure in both DPP3 structures, the "open" and the "semi-open (or "semi-closed"). he main differences (or unique characteristics) between the crystallographic 'open' structure of DPP3 (PDB_ID: 3FVY) and the ‘semi-open’ structure lie in the mutual orientation of two DPP3 domains. That is, the interdomain cleft (gap between domains) is larger in the ‘open’ than in the ‘semi-open’ structure. DPP3 consists of two domains, catalytic and ligand binding domain. Each of these domains has the same structure in both DPP3 structures, the ‘open’ and ‘semi-open(or ‘semi-closed’).
- How many molecular docking servers were used in the study, and what were the criteria for selecting these servers?
Response:
For the results of this study we used three docking servers ClusPro, Haddock and GRAMM-X, details are given in ‘Material and Methods’ section, Paragraph ' Protein Docking’. We also tried some other docking servers, but we could not obtain coherent results with them, so we did not include them in this study. The servers were selected based on literature data and the results of the CAPRI competition.
- Could you explain the significance of adjusting the protonation states of charged residues and histidines to a pH of about 7.5 in the molecular modeling, and how this choice reflects physiological conditions?
Response:
The pH is thought to be about 7.4 under physiological conditions; therefore, to get a more reliable picture of what is happening in the cells, it is important to use this pH in the study. Different pH values would lead to a different distribution of charges in the proteins, i.e., the charges of the amino acid residues are different at different pH values, and consequently the protein–protein interactions are also different.
- What specific criteria or considerations were used to determine the protonation states of histidines, especially those with both Nδ and Nε protonated?
Response:
The ability of His to form either H-bonds or polar interactions with the neighboring residues was the main criterion for deciding whether to protonate Nδ or Nε or both.
- Can you elaborate on the role of the zinc ion in the active site of DPP3 and how it was described using the hybrid bonded/non-bonded model for metallopeptidases?
Response:
The zinc ion bound in the active site of DPP3 has an active role in the enzyme catalysis, as it has been shown that apo DPP3 is enzymatically inactive. The hybrid bonded/non-bonded parameters for the Zn ion used in this study were previously derived by QM /MM calculations in combination with exhausted MD simulations performed for several metalloenzymes whose metal binding site is similar to that in DPP3. The procedure is described in the reference. (24)
- How was the solvation of the system carried out, and why was a truncated octahedron of OPC-water molecules used for this purpose?
Response:
The simulations were performed using periodic boundary conditions, and we assumed that a truncated octahedron was the best choice for studying the dynamics of DPP3 and SH2D3C in their different orientations. Both SH2D3C and DPP3 are mostly globular proteins.
- What was the rationale behind adding Na+ (and/or Cl-) ions to achieve electroneutrality or the desired salt concentration in the system, and how was this accomplished?
Response: When using periodic boundary conditions, it is recommended to use neutral systems to avoid unwanted electron drift across the simulation cell. This is achieved by adding an appropriate number of Na+ (if the system is negatively charged) or Cl- (if the system is positively charged) ions to the system (the ions are added using the modul leap available within the AMBER suit of programs), and positioned to minimize the energy of the system. To obtain the required ionic strength, an appropriate (equal) number of Na+ and Cl- ions are added to the system.
- In the optimization cycles prior to productive MD simulations, why were different restraints applied in each cycle (e.g., constraints on protein and zinc ion, constraints on protein backbone, no additional restraints), and how does this affect system relaxation? What is software utilized?
Response:
In the first cycle, both proteins and the zinc ion were constrained to allow the water molecules to migrate to the optimal positions for the initial conformation of the complex; in the second cycle, only the protein backbone was constrained to allow the side chains of the amino acid residues to orient properly. When the water molecules and the amino acid side chains were in their low-energy orientations, the entire system was relaxed. Optimization was performed with the program sander, and heating, density equilibration, and productive MD simulations were performed with pmemd.cuda. Both programs are available in the AMBER18 suit of programs. The ‘Material and Methods’ section, Paragraph ' Classical MD simulations’ is modified accordingly.
- The author should update the methodology reference, which can be seen PMID: 30602415, 29765814, 33131721, 34551668, 37071766.
Response: The references have been updated with the new references relevant to our study.
- Could you provide more details about the heating process, including the NVT ensemble used and any specific conditions applied to reach 300 K?
Response: We used a Langevin thermostat to heat the system from 0 K to 300 K. The heating was done in three steps. In the first step, the temperature was continuously increased from 0-100 K, in the second from 100-200 K, and in the third from 200-300 K.The NVT ensemble provided a constant volume during heating. The Langevin thermostat maintains the temperature through a modification of Newton's equations of motion according to the equation:
- What was the motivation for using a time step of 0.5 fs during heating simulations and 1 fs during equilibration and production MD simulations?
Response:
During heating, a smaller time step of 0.5 fs was used to allow the system to respond quickly to temperature changes and to avoid unwanted strain and loss of accuracy. Once the system has reached the desired temperature, a time step of 1 fs is sufficient.
- What specific parameters and criteria were used to calculate geometry parameters (RMSD, Rgyration, RMSF), perform hydrogen bonding analysis, and analyze LIE (Ligand Interaction Energy) throughout the MD simulations?
Response:
RMSD (root mean square deviation) is the average difference between equivalent atoms in two structures. During MD simulations a new structure is generated at each time step, so the RMSD calculated for each of these structures and the initial structure gives information about the degree of change of the system during the simulation. In our study, it is calculated according to the following equation:
RMSF (Root-Mean-Square Fluctuation) was used to calculate the average fluctuations of an amino acid residue during MD simulations, equation: .
Rgyr is given by equation
In our study, RMSD, RMSF, and Rgyr were calculated for the heavy backbone atoms on structures generated every 10 ps.
- In RMSD half black trajectory, why?
Response:
The black line in Figure S8 represents the RMSD calculated from the trajectory obtained by simulating model 1. This model was simulated for 400 ns, i.e., shorter than the other models because its calculated binding free energies were less negative that those calculated for models 3,4, and 5.
- What was the purpose of conducting 2 μs of productive classical MD simulations for the predicted DPP3 - SH2D3C complexes?
Response:
Two μs was a simulation time that we could achieve in a reasonable time frame and was still sufficient to obtain an approximate picture of the possible DPP3 -SH2D3C complexes.
- How did you determine the dielectric constant of 2 and the ionic strength of 0.100 mol dm-3 used in the MMGBSA energy calculations?
Response:
These are standard values more
